# PHIP suppresses NuRD to enable the growth of SWI/SNF-mutant cancers

Hayden A. Malone [1,2], Jacquelyn A. Myers[1], Emma G. Gruss [1],
Marc A. Morgan [1], Jake D. Friske [2,3], Tabitha C. McCarty[1], John J. Navarro[1],
Sarah Robinson[3], Rebecca L. Halliburton [1], Sandra J. Kietlinska[1,2],
Francisca N. De Luna Vitorino [4], Baranda S. Hansen [5],
Shondra M. Pruett-Miller [5], Benjamin A. Garcia [4], Martine F. Roussel [2,3],
Janet F. Partridge [1] & Charles W. M. Roberts [1,2] ✉

SWI/SNF chromatin remodeling complexes are perturbed in 20% of all cancers and in several developmental disorders, yet the mechanisms by which these mutations dysregulate transcription and drive disease are poorly understood. To both elucidate these mechanisms and identify vulnerabilities caused by these mutations, we leverage genome-wide CRISPR-Cas9 screening in hundreds of cancer cell lines and identify the chromatin reader protein PHIP as a specific dependency in cancers with broadly disrupted SWI/SNF function. Mechanistically, we reveal that PHIP cooperates with SWI/SNF to facilitate transcriptional activation by ubiquitinating and suppressing subunits of the repressive Nucleosome Remodeling and Deacetylase (NuRD) complex. We demonstrate that loss of SWI/SNF results in NuRD complexes accumulating at promoters where they would otherwise cause widespread transcriptional silencing if not antagonized by PHIP. Collectively, we identify PHIP as a regulator of the interplay between distinct chromatin regulators that function in development and disease and as a targetable vulnerability in cancers with broad SWI/SNF inactivation.

Mammalian development and cell fate specification require precise control of gene programs. Contributing to this control are chromatin regulatory proteins that interact with transcription factors to choreograph the activation and silencing of individual genes. The impact of these protein networks is highlighted by the myriad cancers and other diseases caused by mutations in the genes that encode them[1–4]. Deconvoluting these complex networks will elucidate mechanisms underlying transcription regulation and development, reveal mechanisms driving disease, and potentially enable future therapeutic interventions[5]. Cancer genome sequencing studies have shown that genes encoding subunits of SWI/SNF (switch/sucrose non-fermentable) chromatin remodeling complexes are collectively mutated in more than 20% of cancers[1,2]. SWI/SNF complexes, also known as BAF (BRG1/BRM-associated factor) complexes, use the energy from ATP hydrolysis to mobilize nucleosomes at enhancers and promoters to regulate gene expression. There are three SWI/SNF subfamilies (cBAF, PBAF, and ncBAF), each composed of shared and subcomplex-specific subunits[6]. Several SWI/SNF subunits are encoded by multi-family paralogs, further diversifying the composition and specificity of the complexes. Transcriptional activation mediated by SWI/SNF is counterbalanced by repressive chromatin complexes, such as the Polycomb families[7–9].

[1]Department of Oncology, St. Jude Children's Research Hospital, Memphis, TN, USA. [2]St. Jude Graduate School of Biomedical Sciences, St. Jude Children's Research Hospital, Memphis, TN, USA. [3]Department of Tumor Cell Biology, St. Jude Children's Research Hospital, Memphis, TN, USA. [4]Department of Biochemistry and Molecular Biophysics, Washington University School of Medicine, St. Louis, MO, USA. [5]Center for Advanced Genome Engineering, Department of Cell and Molecular Biology, St. Jude Children's Research Hospital, Memphis, TN, USA. ✉e-mail: charles.roberts@stjude.org

Rhabdoid tumors (RTs) are genomically simple but highly aggressive and lethal cancers driven in all cases by bi-allelic loss of a SWI/SNF subunit (*SMARCB1* in 95% of cases and *SMARCA4* in 5%)[10]. Both *SMARCB1* and *SMARCA4* exhibit bona fide tumor suppressor activity, as germline heterozygous mutations are associated with cancer predisposition in humans[11–13] and knockout in mice results in tumor formation[14,15]. Apart from *SMARCB1* or *SMARCA4* loss, the genomes of these aggressive cancers are diploid and genomically simple. Therefore, RTs constitute a powerful model for identifying chromatin regulators whose function becomes essential after SWI/SNF disruption, which may represent therapeutic vulnerabilities in SWI/SNF-mutant cancers. Small cell carcinomas of the ovary, hypercalcemic type (SCCOHTs), are mechanistically related to RTs. These cancers are driven by biallelic inactivating mutations in the ATPase subunit gene *SMARCA4* and silencing of its mutually-exclusive paralog *SMARCA2*[12,13]. Among the SWI/SNF mutations associated with cancer, those in RTs and SCCOHTs have the most detrimental effects upon SWI/SNF function, as SMARCB1 loss results in degradation of the two major families of SWI/SNF complexes (cBAF and PBAF)[16], and the absence of both SWI/SNF ATPase subunits fully impairs remodeling.

Here, by using genome-scale CRISPR-Cas9 screening of 15 RT cell lines compared to over 1000 other cell lines as part of the Cancer Dependency Map project and the Pediatric Cancer Dependency Accelerator, we identify Pleckstrin homology domain interacting protein (PHIP, or BRWD2/DCAF14/RepID) as a specific dependency in RT cells. PHIP is a chromatin-binding protein that engages nucleosomes through three distinct reader domains[17,18]. Although PHIP has been shown to interact with the CRL4 E3 ubiquitin ligase complex (CUL4A/B, DDB1, and RBX1) and ubiquitinate several targets during DNA replication[19,20], a role in transcriptional regulation has remained elusive. We show that PHIP cooperates with SWI/SNF complexes to activate transcription by ubiquitinating and suppressing subunits of the transcriptionally repressive Nucleosome Remodeling and Deacetylase (NuRD) complex, causing eviction of NuRD from chromatin at sites where PHIP is co-bound. Inactivation of SMARCB1 in RT cells broadly impairs transcriptional activation by SWI/SNF, and we demonstrate that this results in enhanced recruitment of NuRD complexes to promoters that would otherwise result in transcriptional collapse. In this setting, PHIP becomes essential to suppress NuRD-mediated silencing and rescue transcription at target genes to enable cell proliferation. Collectively, our findings identify PHIP as a critical regulator of the interplay between two antagonistic chromatin regulatory complexes and as a potential therapeutic target in select families of SWI/SNF-mutant cancers. Of relevance, PHIP contains a uniquely targetable bromodomain, and efforts to develop small-molecule inhibitors have begun[21,22].

## Results

### Cancers with broad disruption of SWI/SNF are sensitive to PHIP inactivation

To identify vulnerabilities caused by SWI/SNF mutations in cancer, we used data from genome-wide CRISPR screens performed in more than 1000 cancer cell lines through the Cancer Dependency Map Project[23], including 15 SMARCB1-deficient RT cell lines. We performed a two-class comparison of dependency scores in RT cell lines versus all other cancers and identified PHIP as a selective dependency ($P = 1.14E − 9$) (Fig. 1a). Each of the four gRNAs in the Avana library that target PHIP were selectively disadvantageous to RT and SCCOHT cell lines, indicating that the effect was on-target (Supplementary Fig. 1A). This vulnerability scored as being more significant than other mechanistically informed vulnerabilities that are being pursued in clinical trials (MDM4[24], BRD9[25,26]) and one target (EZH2[27]) that has received FDA approval in SMARCB1-mutant cancers (Supplementary Fig. 1B). Although PHIP has been described as a putative and potentially targetable chromatin-associated protein[17,18,21,22], its function was still unclear.

The dependency upon PHIP extended to cancers with mutations that broadly impair the function of multiple SWI/SNF subfamilies, such as small-cell carcinomas of the ovary, hypercalcemic type (SCCOHTs) ($P = 1.39E − 10$) (Fig. 1b). In contrast, more limited impairment of SWI/SNF function, such as in cancers with mutations that affect only one member of a paralog pair (e.g., a mutation in *ARID1A* but not *ARID1B*) or only one SWI/SNF subfamily (e.g., a mutation in the PBAF-specific subunit gene *PBRM1*), do not result in enhanced sensitivity to PHIP loss (Supplementary Fig. 1C).

We next sought to further validate PHIP as a specific vulnerability. We first performed a CRISPR-mediated frameshift fitness assay, CelFi[28], using additional gRNAs that target PHIP at two more distinct loci (Fig. 1c). Because vulnerabilities can sometimes reflect the cell of origin rather than the mechanism underlying oncogenic transformation, we used cancer cell lines from multiple tissues (G401: a *SMARCB1*$^{−/−}$ kidney rhabdoid tumor; CHLA266: a *SMARCB1*$^{−/−}$ brain atypical teratoid rhabdoid tumor (AT/RT); BIN67: a *SMARCA2/4*$^{−/−}$ SCCOHT). Selection against damaging PHIP mutations was observed in these SWI/SNF-mutant cancer cell lines from distinct tissues but not in a control cell line that lacked SWI/SNF mutations (Fig. 1d). Additionally, growth assays demonstrated impaired proliferation of multiple RT cell lines originating from distinct tissues after PHIP inactivation using shRNAs or sgRNAs (Supplementary Fig. 1D–G). Further confirming that the effects were on-target, the anti-proliferation effects of sgRNAs targeting PHIP were rescued by over-expressing an HA-PHIP construct that was not recognized by the guides (Supplementary Fig. 1H).

To directly test whether the absence of SMARCB1 caused sensitivity to PHIP loss, we knocked out SMARCB1 in 293 T cells (which are wildtype for SWI/SNF) and evaluated their dependency upon PHIP (Supplementary Fig. 1I). Only 293 T cells deficient in SMARCB1 were sensitive to PHIP knockout (Fig. 1e).

### PHIP activates transcription

To investigate the essential function of PHIP in RTs, we first asked where it bound to chromatin. ChIP-seq revealed that PHIP co-localizes with histone posttranslational modifications (PTMs) that mark the promoters of actively transcribed genes (H3K4me3, H3K27ac, H3K14ac) and that PHIP binds primarily near transcription start sites (TSSs) in RTs (Fig. 2a, Supplementary Fig. 2A, B). In contrast, PHIP did not co-localize with repressive sites marked by H3K27me3 (Supplementary Fig. 2C).

Given its localization to gene regulatory elements, we asked whether PHIP regulated transcription in RT cells. By RNA-seq, we observed an enrichment of downregulated genes after PHIP knockdown and an enrichment of upregulated genes upon PHIP overexpression, suggesting that PHIP has an activating role (Fig. 2b, c, Supplementary Fig. 2D). Transcriptional changes were validated by independent siRNAs (Supplementary Fig. 2E, F). We next asked whether PHIP binding is enriched at differentially expressed genes by using Binding and Expression Target Analysis (BETA)[29]. This revealed that PHIP binding is significantly enriched at genes that are downregulated upon PHIP inactivation ($P = 2.72E − 08$) (Supplementary Fig. 2G), supporting a model wherein PHIP binds to promoters of genes and directly enhances their expression. Further supporting on-target activity of PHIP as a transcriptional activator in RT cells, we identified 2077 direct target genes that are upregulated when PHIP is over-expressed, downregulated when PHIP is knocked out, and rescued upon PHIP rescue (Fig. 2d).

We next investigated whether PHIP altered the chromatin landscape in RT. We found that PHIP is critical for preserving acetylation of several histone posttranslational modifications (PTMs) associated with gene activation, including H4ac, H3K27ac, and H3K14ac (Fig. 2e,

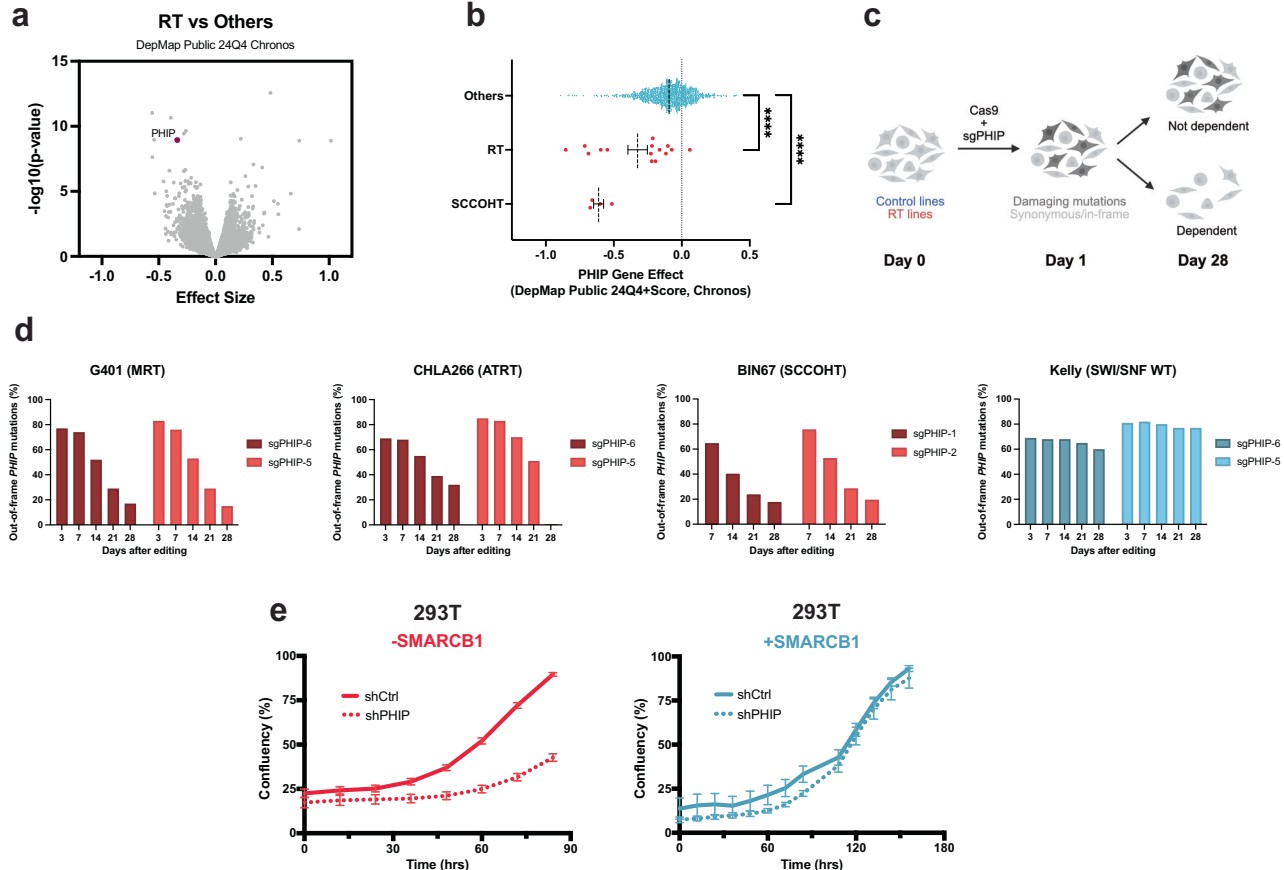

**Fig. 1 | Cancers with broad disruption of SWI/SNF are sensitive to PHIP inactivation. a** Two-class comparison of Chronos dependency scores of genes in RT cell lines (*n* = 15) compared to all other cell lines (*n* = 1086) in DepMap (24Q4 Chronos release). Each point represents a gene, and more negative scores indicate selective dependency upon that gene. P-values were calculated using a two-tailed Benjamini–Hochberg-corrected Student's *t*-test. **b** Scatter plot comparing dependency scores for PHIP in RT (*n* = 15) and SCCOHT (*n* = 4) versus all other cell lines (*n* = 1082) in DepMap (24Q4 Chronos release). Each point represents a cell line, and more negative scores indicate enhanced sensitivity to PHIP inactivation. Statistical analysis was performed using a two-tailed Benjamini–Hochberg-corrected Student's *t*-test. *P* = 1.14E − 9 for RT lines; *P* = 1.39E − 10 for SCCOHT lines. The center line represents the mean, and the error bars represent the standard error of the mean (SEM). **c** Schematic of CelFi assay used to validate specific dependency

upon PHIP. Created in BioRender. Malone, H. (2026) https://BioRender.com/y6pmjrh. **d** Results from CelFi indel toxicity assay in which CRISPR-mediated editing was used to create out-of-frame alleles in PHIP. Bar charts show the percentage of out-of-frame PHIP alleles over time in *SMARCB1*[−/−] RT cell lines originating from the kidney (G401, red) and brain (CHLA266, red) and in SCCOHT cells lacking SMARCA4 and SMARCA2 (BIN67, red), compared to a control neuroblastoma cell line that lacks mutations in SWI/SNF subunits (Kelly, blue). Sequencing was performed at days 3, 7, 14, 21, and 28 after editing. **e** Cell confluency over time in 293 T cells expressing SMARCB1 (blue) or SMARCB1-null 293 T (red) after transduction with scramble shRNAs (solid line) or shRNAs targeting PHIP (dashed line). Data are mean confluency measurements from *n* = 8 technical replicates, and error bars represent SEMs. Source data are provided as a Source Data file.

Supplementary Fig. H, I). We then asked where histone acetylation was lost when PHIP was inactivated. ChIP-seq revealed that deacetylation occurs primarily at active promoters where PHIP is bound (Fig. 2f), suggesting that PHIP locally preserves the acetylation of the loci to which it binds. PHIP loss did not appear to affect H3K4me3, which is found at the promoters of active genes. Genetic rescue experiments confirmed that control of H3K27ac by PHIP is indeed on-target (Fig. 2g).

Finally, we asked whether the targets of PHIP could explain its critical role in RTs. We found that PHIP activates the expression of genes regulating cell proliferation, which is consistent with the observation that PHIP inactivation results in phenotypic evidence of cell cycle arrest (Supplementary Fig. 2J, K). PHIP's role as a transcriptional activator is conserved in additional RT (A204), AT/RT (CHLA266), and SCCOHT (BIN67) cell lines, where it also enhances pro-proliferative expression programs (Supplementary Fig. 2L, M). These data implicated PHIP as an essential chromatin regulator in RTs, but the mechanism by which PHIP preserved histone acetylation and gene activation remained elusive.

## PHIP recruits E3 ligases to chromatin and ubiquitinates NuRD

We next sought to elucidate the mechanism through which PHIP, which lacks acetyltransferase activity, preserves histone acetylation in RT cells. PHIP has been reported to recruit CRL4 E3 ubiquitin ligase complexes to chromatin to ubiquitinate target substrates[19,20]. This led us to hypothesize that PHIP activated transcription by ubiquitinating and suppressing the function of repressive chromatin regulators such as histone deacetylases.

Consistent with prior studies[18,19], we found that PHIP both interacts with the CRL4 complex and is required for recruiting CRL4 subunits CUL4A, CUL4B, and DDB1 to chromatin (Fig. 3a, Supplementary Fig. 3A−D). To determine whether cooperation of PHIP and CRL4 was critical for RT cells, we used a mutant version of PHIP that cannot interact with CRL4 (ΔH-PHIP[30], Fig. 3b). We confirmed that this mutation eliminates interactions between CRL4 and PHIP (Fig. 3c) and that ΔH-PHIP fails to recruit CRL4 to chromatin (Fig. 3d). We next asked whether the essentiality of PHIP in RT was dependent on its cooperation with CRL4. ΔH-PHIP could not rescue growth defects after PHIP knockout (Supplementary Fig. 3E), suggesting that coordination with

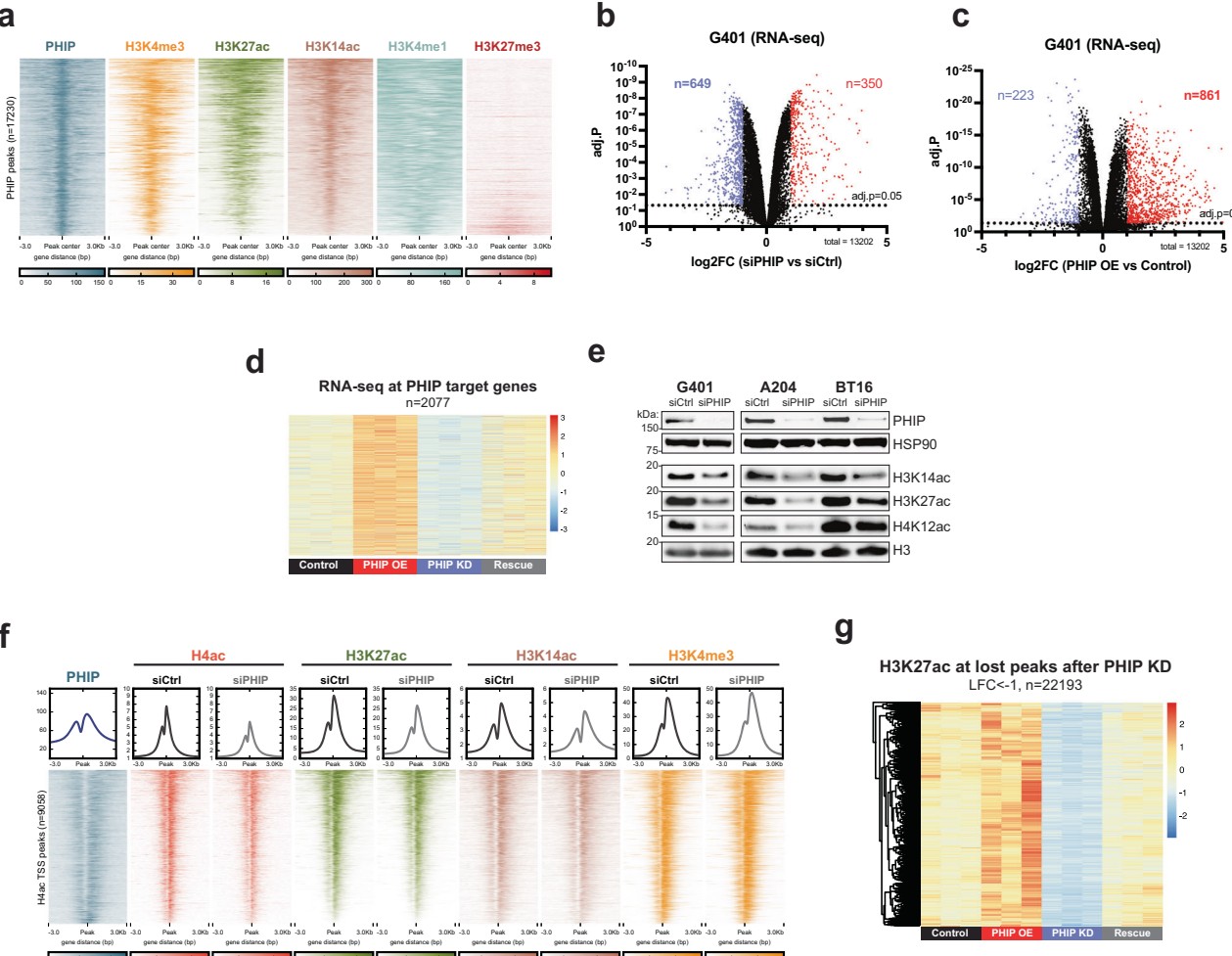

**Fig. 2 | PHIP activates transcription. a** Heatmap of ChIP-seq signal for PHIP (blue, $n = 3$), and posttranslational modifications (PTMs) marking active promoters, i.e., H3K4me3 (orange, $n = 3$), H3K27ac (green, $n = 3$), and H3K14ac (salmon, $n = 3$) at PHIP peaks ($n = 17{,}230$) in G401 cells, sorted by H3K27ac intensity. Also shown are the enhancer marker H3K4me1 (cyan, $n = 2$) and the repressive/bivalent marker H3K27me3 (red). Volcano plot of differentially expressed genes after knocking down (**b**) or overexpressing (**c**) PHIP in G401 RT cells ($n = 3$). Statistical analysis was performed using a two-sided empirical Bayes moderated $t$-test using the limma package. The dashed line indicates the adjusted $P = 0.05$. For significantly down-regulated genes (blue): adjusted $P < 0.05$, log2FC $< -1$; for significantly upregu-lated genes (red): adjusted $P < 0.05$, log2FC $> 1$. **d** Heatmap of RNA-seq signal for PHIP target genes ($n = 2077$) in control (black), PHIP-overexpressing (red), PHIP-knockdown (purple), and PHIP-rescue (gray) G401 cells. **e** Western blots showing acetylation of H3K14, H3K27, and H4K12 in histone lysates from siCtrl-treated and siPHIP-treated RT cell lines (G401, A204, and BT16). Blots are representative of $n = 3$ biological replicates for all cell lines. H3 and HSP90 are loading controls. **f** Evaluation of the effects of PHIP knockdown on histone acetylation. Heatmap of ChIP-seq signal for PHIP (blue, $n = 3$) in G401 cells and for H4ac (cherry, $n = 3$), H3K27ac (green, $n = 3$), H3K14ac (salmon, $n = 3$), and H3K4me3 (orange, $n = 3$) in siCtrl-treated and siPHIP-treated G401 cells at active TSSs (H4ac + , $n = 9058$). Sorted by H3K27ac. **g** Heatmap of H3K27ac ChIP-seq signal at sites where H3K27ac is lost after PHIP loss ($n = 22{,}193$) in control (black), PHIP-overexpressing (red), PHIP-knockdown (purple), and PHIP-rescue (gray) G401 cells. Source data are provided as a Source Data file.

CRL4 is critical for PHIP's essential function as a transcriptional acti-vator in RTs.

As it was still unknown where on chromatin PHIP recruited CRL4 and whether this affected transcription, we performed ChIP-seq for CRL4 subunit DDB1. There was strong co-localization of PHIP and DDB1, which was substantially enriched at active promoters corre-sponding to the +1 nucleosome directly downstream of the nucleosome-free region (Fig. 3e, Supplementary Fig. 3F). Knockdown of PHIP resulted in reduced binding of DDB1 (Supplementary Fig. 3G). To investigate whether the impaired recruitment of CRL4 affected transcription, we used BETA to determine whether loci to which PHIP recruits CRL4 are enriched at differentially expressed genes after PHIP knockdown. Sites that lose DDB1 binding after PHIP loss are strongly enriched at downregulated genes ($P = 3.1E - 20$) (Supplementary Fig. 3H). These findings suggest that PHIP recruits CRL4 to promoters of genes to facilitate their expression.

To search for substrates targeted by CRL4-PHIP on chromatin, we performed immunoprecipitation mass spectrometry (IP-MS) and detected interactions between PHIP and 10 distinct subunits of the Nucleosome Remodeling and Deacetylase (NuRD) complex (Fig. 3f). This complex was compelling as a potential target of PHIP, as it has deacetylase activity and has been reported to behave antagonistically to SWI/SNF complexes in development and disease. NuRD complexes are multi-subunit chromatin-modifying complexes that repress tran-scription by uniting two enzymatic activities: histone deacetylation (by HDAC1/2) and nucleosome remodeling (by CHD3/4/5). Structurally, NuRD complexes are assembled onto chromatin as two modules capable of distinct enzymatic functions: a deacetylase module con-sisting of HDAC1/2, RBBP4/7, and MTA1/2/3 and a remodeling module consisting of CHD3/4/5, GATAD2A/B, and CDK2AP1[31–33]. MBD2/3 bridges the two modules to constitute complete NuRD complexes with remodeling and deacetylase activity.

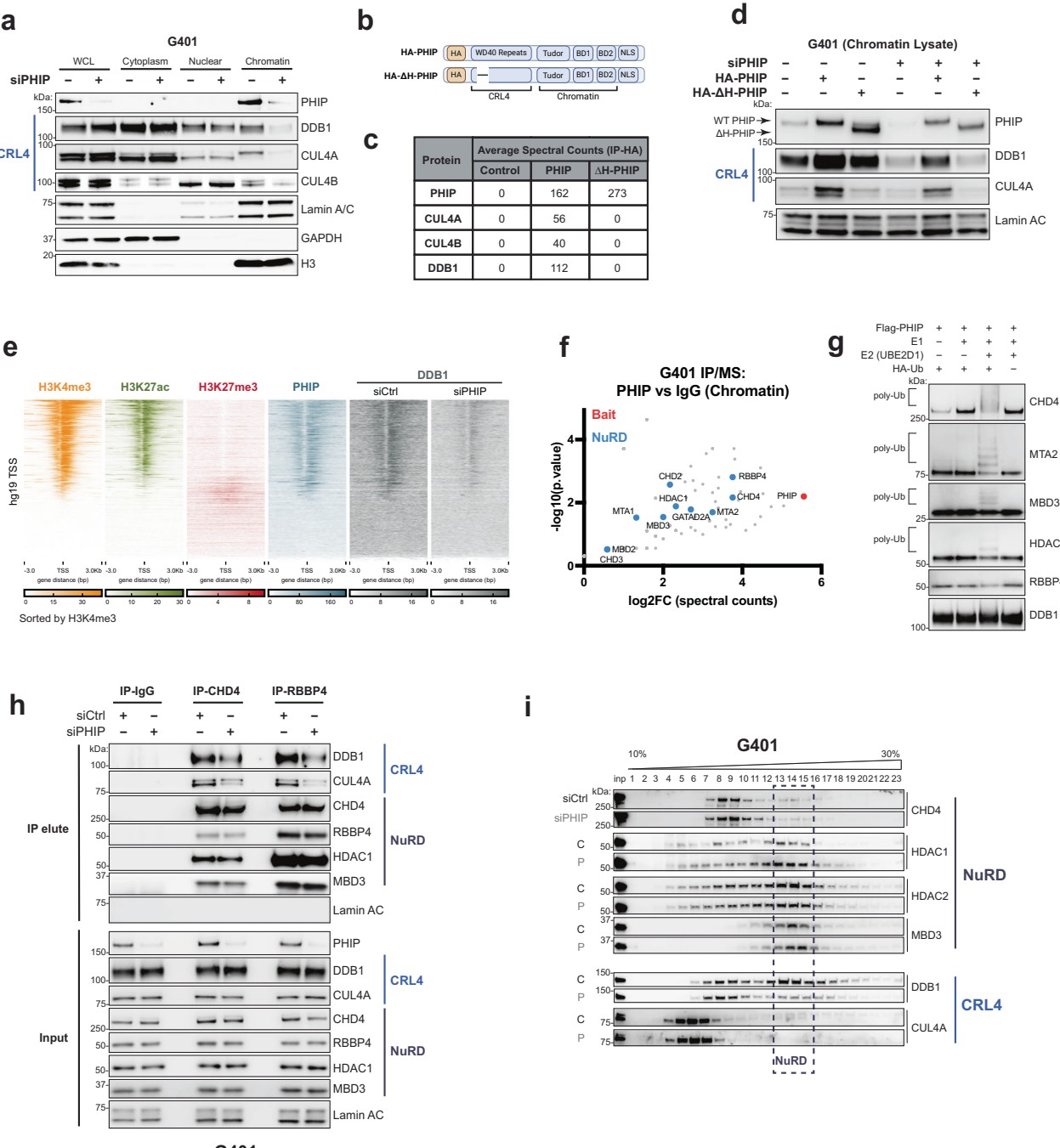

**Fig. 3 | PHIP recruits E3 ligases to chromatin and ubiquitinates NuRD. a** Western blots (WBs) of whole-cell lysate (WCL), and cytoplasmic, nuclear, and chromatin fractions with and without PHIP knockdown in G401 cells. GAPDH: cytoplasmic fraction loading control. Lamin A/C: nuclear soluble loading control. H3: chromatin loading control. Blots are representative of *n* = 3 biological replicates. **b** Schematic of HA-PHIP and HA-ΔH-PHIP constructs. **c** Spectral counts for PHIP and members of the CRL4 complex from IP-MS after affinity purification of HA-PHIP or HA-ΔH-PHIP in G401 cells (averaged from *n* = 2 biological replicates). **d** WBs of chromatin fractions in control or PHIP-knockout cells after inducing expression of HA-PHIP or HA-ΔH-PHIP. Lamin A/C: loading control. Blots are representative of *n* = 3 biological replicates. **e** Heatmap showing ChIP-seq coverage (averaged from *n* = 3 biological reps) for H3K4me3 (orange), H3K27ac (green), H3K27me3 (red), and PHIP (blue) in G401 RT cells, and DDB1 (gray) in siCtrl- and siPHIP-treated G401 at all annotated TSSs (hg19). Sorted by H3K4me3 signal strength. **f** Scatter plot displaying proteins enriched by PHIP pulldown compared to an IgG negative control in G401 RT cells as measured by mass spectrometry (*n* = 2 biological replicates).

Statistical analysis was performed using a two-tailed Student's *t* test. Each point represents a unique protein. Results were filtered for chromatin-bound proteins (using the Chromatin GO:0000785 gene list). PHIP and NuRD subunits are shown in red and blue, respectively. The x-axis represents the enrichment of PHIP interactors (log2FC of PHIP IP vs IgG), and the y-axis represents significance [−log10 (*P* value)]. **g** WBs of products from in vitro ubiquitination assays using FLAG-purified PHIP from 293 T cells. Data are representative of *n* = 3 biological replicates. **h** Co-immunoprecipitation of NuRD subunits (CHD4 and RBBP4) in siCtrl-treated and siPHIP-treated G401 cells. Lamin A/C is a negative control, and the input shown is 2% of the lysate used for the co-IP. Data are representative of *n* = 3 biological replicates. **i** Glycerol gradient (10–30%) density sedimentation of nuclear lysates to determine whether NuRD complexes and CRL4 complexes are detected in similar fractions and whether their co-migration is affected by PHIP loss in G401 cells. Blots are representative of *n* = 3 biological replicates. Source data are provided as a Source Data file.

To evaluate whether subunits of the NuRD complex were targets of PHIP-mediated ubiquitination, we performed in vitro ubiquitination assays. We found that FLAG-purified CRL4-PHIP exhibits robust ubiquitin ligase activity toward co-purified CHD4 and MTA2, with lesser activity toward MBD3 and HDAC1, and minimal activity toward RBBP4 (Fig. 3g). FLAG-enriched CRL4-PHIP directly recognized and ubiquitinated recombinant CHD4 added to the assay, but not recombinant HDAC1 (Supplementary Fig. 3I). This suggests that CHD4 is directly recognized and ubiquitinated by PHIP, whereas HDAC1, which is also a member of several other chromatin regulatory complexes, is ubiquitinated by PHIP only when it is structurally associated with NuRD. However, PHIP probably binds to multiple NuRD subunits, as knockdown of CHD4 does not eliminate interactions between PHIP and other NuRD members in RT cells (Supplementary Fig. 3J).

This observation in vitro led us to investigate whether CRL4-PHIP targeted NuRD complexes for ubiquitination in RT cells. Co-immunoprecipitations revealed that the binding of CRL4 subunits (DDB1 and CUL4A) to NuRD subunits (CHD4 and RBBP4) is impaired in the absence of PHIP (Fig. 3h). Additionally, glycerol gradient fractionations revealed that CRL4 subunits DDB1 and CUL4A no longer co-migrate with NuRD in the absence of PHIP (Fig. 3i). Together, these findings suggest that PHIP facilitates the recognition and ubiquitination of NuRD complexes by CRL4 in RT cells.

## PHIP suppresses silencing by NuRD complexes at promoters

Next, we investigated whether PHIP influenced NuRD binding and function in RTs. Although overexpressing PHIP did not affect overall cellular levels of NuRD, it led to reduced levels of several NuRD subunits on chromatin (Fig. 4a). In line with impaired NuRD function on chromatin, levels of histone acetylation rose (Fig. 4a).

To determine whether PHIP coordinates with CRL4 to suppress NuRD, we overexpressed the ΔH-PHIP mutant that cannot recruit CRL4. Unlike full-length PHIP, ΔH-PHIP expression did not diminish NuRD abundance on chromatin (Fig. 4b). Next, we chemically inhibited CRL4 using neddylation inhibitors (MLN-4924) and found that this prevented PHIP-mediated suppression of NuRD on chromatin (Supplementary Fig. 4A). Together, these results indicate that PHIP-mediated suppression of NuRD is cullin-dependent.

Ubiquitination of chromatin-associated proteins can suppress their function in distinct ways, including targeting them for proteasomal degradation or removing them from chromatin independent of degradation. To distinguish between these possibilities, we treated the cells with the proteasome inhibitor MG132 but found that MG132 treatment did not lead to detectable rescue (Supplementary Fig. 4B). This may indicate that PHIP ubiquitinates and suppresses NuRD function through chromatin eviction alone independent of proteasomal degradation, though it is important to note that proteasome inhibition can stimulate autophagy as a backup degradation method for certain proteins[34–36]. These results, together with our in vitro ubiquitination studies, suggest that PHIP recruits CRL4 to mediate the ubiquitination of NuRD complexes and their removal from chromatin in RT cells, thus preserving histone acetylation. We next investigated where these events occurred on chromatin and how this influenced gene expression.

First, we asked how chromatin binding of PHIP related to that of NuRD. ChIP-seq revealed two classes of NuRD binding sites: there is strong binding at loci with chromatin modifications consistent with active promoters and weaker binding at sites lacking histone acetylation (Fig. 4c). We found that PHIP flanks NuRD exclusively at the strongly bound peaks resembling active promoters (Fig. 4c). We next evaluated the effect of PHIP on NuRD binding, initially focusing on CHD4 because PHIP displays the strongest ubiquitination activity towards this subunit in vitro. Upon PHIP loss, CHD4 accumulated at promoters, corresponding to reduced acetylation of histone residues that are known targets of NuRD-mediated deacetylation (Fig. 4d, e).

Sites where CHD4 accumulated corresponded to PHIP binding sites before knockdown, whereas sites with unchanged CHD4 binding were not previously bound by PHIP (Fig. 4f). To investigate whether PHIP influenced transcription by targeting CHD4, we used BETA to determine whether sites that gained CHD4 were enriched at differentially expressed genes. This analysis revealed that sites that gain CHD4 binding are enriched at downregulated genes after PHIP loss ($P = 1.77E − 9$) (Supplementary Fig. 4C).

We next evaluated the effects of PHIP loss on NuRD subunits for which PHIP displayed weaker ubiquitination activity (HDAC1, MTA2, and RBBP4). ChIP-seq revealed that these subunits are already present at the sites that gain CHD4 and that they are minimally affected by PHIP knockdown (Supplementary Fig. 4D, E). As remodeling by CHD4 enhances deacetylation of nucleosomes by HDACs within NuRD[37–39], we asked whether the loss of histone acetylation caused by PHIP knockdown was dependent upon CHD4. This was indeed the case, as knockdown of CHD4 rescued the acetylation of several histone residues to near-wildtype levels in the absence of PHIP (Fig. 4g). Together, these data identify a role for PHIP as a transcriptional regulator that suppresses NuRD-mediated silencing at promoters by targeting CHD4.

Finally, to evaluate whether this activity of PHIP was conserved in other cancers lacking SWI/SNF function, we overexpressed PHIP in the BIN67 SCCOHT cell line. Similar to the effects observed in RT cells, overexpressing PHIP in a SCCOHT cell line resulted in enhanced eviction of NuRD subunits from chromatin and impaired histone deacetylation (Fig. 4h). Together, these findings identify a role for PHIP as a transcriptional activator that ubiquitinates and suppresses the NuRD complex, and they demonstrate that this role is conserved in distinct cancers with broadly impaired SWI/SNF function.

## SWI/SNF loss stimulates suppression of NuRD by PHIP

To investigate why PHIP was specifically essential in SWI/SNF-mutant cancers, we first asked whether the localization or function of NuRD complexes was affected by SWI/SNF mutations. SWI/SNF complexes usually localize to both enhancers and promoters. Although inactivating mutations in SMARCB1 or SMARCA4 result in marked impairment of enhancer function[40,41], the function of many promoters is rescued by other chromatin regulators[42]. This results in a reliance on promoter-centric gene activation, a common consequence of SWI/SNF mutations.

To investigate how NuRD function changed upon SWI/SNF loss, we rescued SMARCB1 expression in RT cells. Induction of SMARCB1 did not affect the abundance of NuRD subunits bound to chromatin (Fig. 5a). However, ChIP-seq revealed a broad redistribution of NuRD in the absence of SMARCB1, with differential effects at enhancers and promoters. Upon SMARCB1 rescue, CHD4 accumulated at enhancers that were restored (Fig. 5b, c). At promoters, however, the opposite occurred: CHD4 binding was reduced, indicating that NuRD accumulates at promoters in the absence of SMARCB1 (Fig. 5c, d). Together, these data support a model wherein repressive NuRD complexes are recruited to counterbalance SWI/SNF-driven activation, thereby enabling a tunable system of gene expression. When SWI/SNF-mediated activation of enhancers is impaired, NuRD complexes redistribute to promoters. In this setting, PHIP becomes essential to oppose aberrant silencing by NuRD and to preserve promoter-driven expression in cells with defective enhancer function.

To evaluate whether degradation of NuRD by CRL4-PHIP was specific to RT and SCCOHT lineages or reflected a broader consequence of SWI/SNF inactivation, we used 293 T cells with intact SWI/SNF complexes. Overexpressing PHIP did not affect NuRD turnover on chromatin in 293 T cells (Fig. 5e). However, after the SWI/SNF ATPase subunits SMARCA4 and SMARCA2 were eliminated via treatment with a PROTAC (AU-15330)[43], PHIP overexpression enhanced NuRD eviction from chromatin (Fig. 5e). Together, these findings demonstrate that cancer-associated mutations that broadly impair SWI/SNF function or

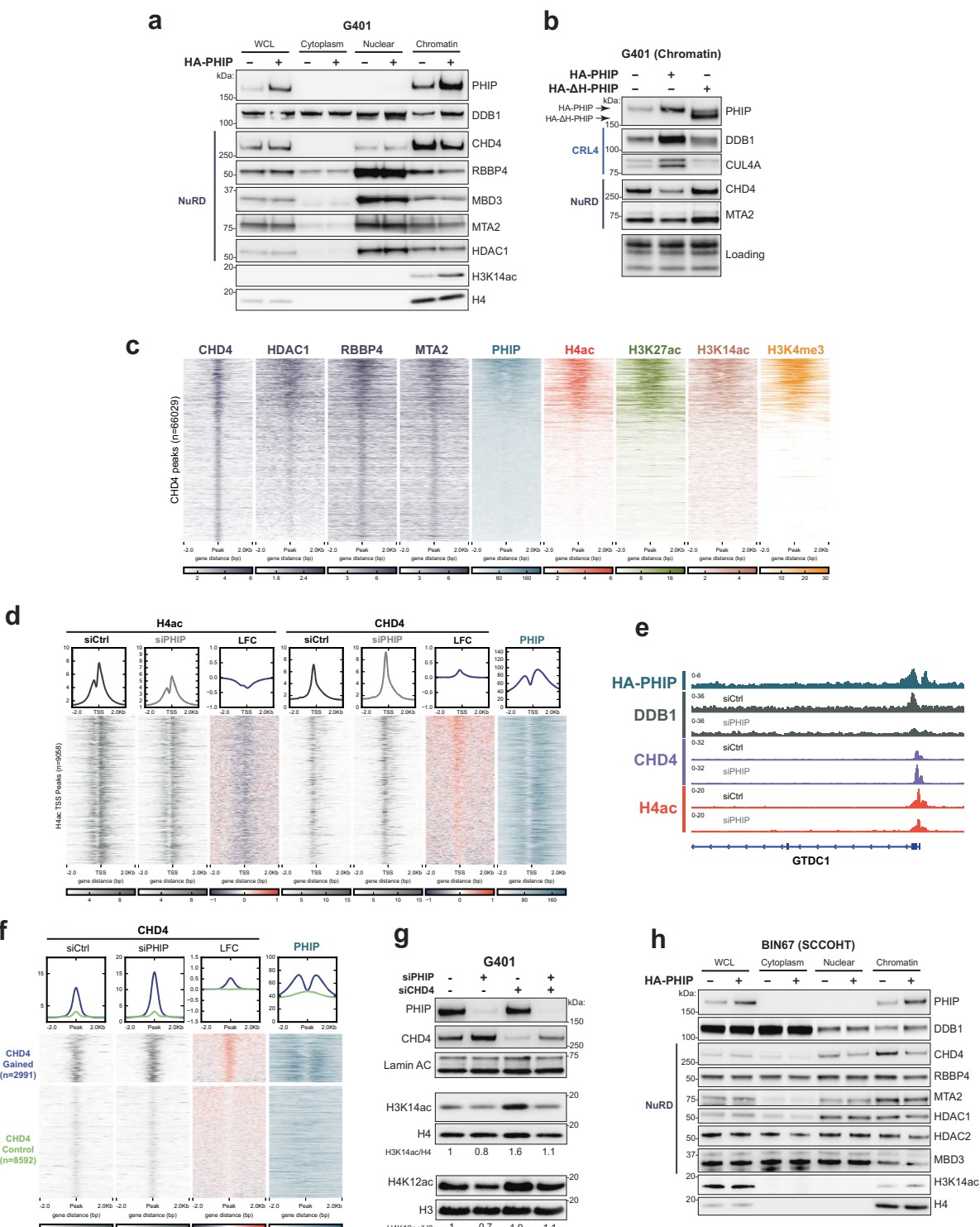

drugs targeting SWI/SNF may stimulate the removal of NuRD from chromatin by CRL4-PHIP.

## Patient-derived in vivo models of RTs specifically depend upon PHIP

To evaluate whether RTs were dependent upon PHIP in vivo, we used recently developed patient-derived orthotopic xenograft (PDOX) models of pediatric brain RTs (termed AT/RTs) that are genetically, transcriptionally, and histologically faithful to the tumors from which they were derived (Fig. 6a)[44,45]. To first evaluate whether PHIP was required for these patient-derived models in vitro, we performed flow-based fitness assays in $SMARCB1^{-/-}$ AT/RT tumor organoids representing distinct disease subtypes, SJATRT041800 (SHH) and SJATRT059003 (MYC), and in a control medulloblastoma tumor organoid line that lacks SWI/SNF mutations, SJMB016874. Consistent with our observations in cell lines, PHIP loss was poorly tolerated in patient-derived AT/RT tumor organoids with impaired SWI/SNF function but had little effect in control organoids (Fig. 6b, Supplementary Fig. 5A, B).

We next orthotopically xenografted AT/RT tumor organoids treated with non-targeting ($n$ = 9) or PHIP-targeting ($n$ = 10) sgRNAs (Fig. 6c) into the cortices of naïve mice and monitored disease

**Fig. 4 | PHIP suppresses silencing by NuRD complexes at promoters. a** Western blots (WBs) of whole-cell lysate (WCL) and cytoplasmic, nuclear, and chromatin fractions after PHIP overexpression in G401 cells. H4: WCL and chromatin loading control. Blot is representative of $n$ = 3 biological replicates. **b** WBs of chromatin lysates after overexpressing full-length or ΔH-PHIP in G401 cells. Total protein stain of histones included as a loading control. Blot is representative of $n$ = 2 biological replicates. **c** Heatmap showing ChIP-seq coverage for NuRD subunits CHD4, HDAC1, RBBP4, and MTA2 (purple, $n$ = 3), PHIP (blue, $n$ = 3), H4ac (red, $n$ = 3), H3K27ac (green, $n$ = 3), H3K14ac (salmon, $n$ = 3), and H3K4me3 (orange, $n$ = 3) at CHD4 reproducible peaks ($n$ = 66,029). Sorted by PHIP signal strength. **d** Heatmap showing ChIP-seq coverage for H4ac and CHD4 in siCtrl-treated and siPHIP-treated G401 cells (gray, $n$ = 3), the log2 fold change (LFC) of the H4ac and CHD4 signal in siCtrl versus siPHIP conditions (navy/red), and PHIP (blue, $n$ = 3) at TSS-associated H4ac peaks ($n$ = 9058). Sorted by LFC CHD4. **e** Example promoter showing the effect of PHIP knockdown upon the CRL4 subunit DDB1 (gray), the NuRD subunit CHD4 (purple), and acetylated histone 4 (red) in G401 cells. **f** Heatmap showing ChIP-seq coverage for CHD4 in siCtrl and siPHIP conditions (gray, $n$ = 3), the log2 fold-change (LFC) of the CHD4 signal in siCtrl versus siPHIP conditions (navy/red), and PHIP (blue, $n$ = 3) at sites where CHD4 is gained ($n$ = 2991, blue) or unchanged ($n$ = 8592, green) after PHIP knockdown in G401 cells. Sorted by LFC CHD4. **g** WBs of chromatin lysates illustrating how histone acetylation changes in the absence of PHIP and/or CHD4 in G401 cells. Lamin A/C: loading control. H3: loading control for H4K12ac. H4: loading control for H3K14ac. Band intensity was quantified and normalized to the loading control, then calculated relative to the siCtrl condition. Blots are representative of $n$ = 3 biological replicates. **h** WBs of WCL and cytoplasmic, nuclear, and chromatin fractions after PHIP overexpression in BIN67 SCCOHT cells. H4: chromatin loading control. Blots are representative of $n$ = 2 biological replicates. Source data are provided as a Source Data file.

progression over time. PHIP inactivation significantly extended survival ($P$ = 0.0017), with a median survival of 37 days in the non-targeting control cohort (95% CI: 24–44 days), compared with 69 days in the PHIP-knockout cohort (95% CI: 37–108 days) (Fig. 6d). As tumors eventually arose in a subset of the PHIP-knockout cohort, this raised the question of whether these tumors arose from cells that had escaped PHIP inactivation. Immunoblots of tumor samples revealed that PHIP expression was partially restored in the lethal tumors (Fig. 6e). Consistent with this, we observed selection against PHIP loss over time in the pools of cells used for implantation that were simultaneously grown in culture in vitro for the duration of the survival study (Supplementary Fig. 5C, D). These findings demonstrate that PHIP is specifically essential in patient-derived in vivo models of RT.

## Discussion

The key roles of the regulatory networks controlling gene expression during development and homeostasis are highlighted by the many cancers and other diseases resulting from mutations of genes encoding transcription factors and chromatin regulatory proteins. SWI/SNF and NuRD complexes oppose each other in transcriptional regulation[46–48], but the mechanisms underlying this antagonism have been poorly understood. Here, we have identified a function for PHIP in transcriptional activation by suppressing the function of repressive NuRD complexes to promote gene expression (Fig. 7). The cooperative functionality of PHIP and SWI/SNF is highlighted by PHIP becoming a specific vulnerability in cancers with mutations that substantially impair SWI/SNF complexes.

The strong dependence upon PHIP in RTs and SCCOHTs may reflect the broad disruption of SWI/SNF complexes that result, respectively, from SMARCB1 loss (inactivating the cBAF and PBAF families of SWI/SNF complexes) in RTs and the combined absence of the SWI/SNF ATPase paralogs in SCCOHTs. In contrast, mutations that affect a single SWI/SNF subfamily or in genes encoding one subunit of a SWI/SNF paralog pair do not impair SWI/SNF as strongly[49,50] and, correspondingly, do not display dependence upon PHIP. This suggests that aberrant NuRD silencing and PHIP dependency are consequences of a broad impairment of SWI/SNF function. The role of PHIP may be related to *PHF6*, a gene we previously identified as a dependency in RTs and SCCOHTs[51]. PHF6 has phenotypic and mechanistic overlap with PHIP in disease and development, and the two proteins interact and co-localize on chromatin[52]. PHF6 had previously been shown to bind to NuRD subunit RBBP4, but the mechanism by which they regulate chromatin and transcription had been unclear[53]. Our identification of PHIP as a regulator of the transcriptional balance between SWI/SNF and NuRD complexes provides insight into the mechanisms that underlie chromatin-mediated control of cell fate in highly disease-relevant contexts.

The identification of antagonism between SWI/SNF and repressive Polycomb complexes ultimately led to the FDA approval of an enzymatic inhibitor of Polycomb subunit EZH2 for treating SMARCB1-mutant cancers[5]. However, the clinical impact has been modest due to the emergence of resistance[54]. Our identification of PHIP as a modulator of NuRD–SWI/SNF antagonism opens the door to a new avenue of potential therapeutic interventions in cancers with SMARCB1 loss and those lacking SMARCA4 and SMARCA2, as efforts to develop small-molecule inhibitors of PHIP continue[21,22]. While therapeutic windows can only be fully evaluated in clinical trials, our finding that genetic inactivation of PHIP has a minimal effect in over 1000 other cell lines in DepMap, complemented by our finding that suppression of NuRD by PHIP is stimulated by SWI/SNF inactivation, raises the possibility of a robust on-target therapeutic window.

## Methods
### Cell culture
G401 (ATCC-CRL1441), A204 (ATCC-HTB-82), and HEK293T (ATCC-CRL-3216) cell lines were obtained from ATCC. Kelly cells were provided by Adam Durbin at St. Jude Children's Research Hospital. BIN67 cells were provided by Bernard E. Weissman at the University of North Carolina. The CHLA-266 cell line was provided by the Children's Oncology Group. BT16 cells were obtained through a material transfer agreement from C. D. James. TTC549 cells were provided by Timothy Triche at UCLA. All cells were maintained in culture at 37 °C, 95% humidity, and 5% $CO_2$. The identities of all cell lines were validated using STR profiling, and cells were regularly tested for mycoplasma by PCR (Genlantis). G401 and A204 cells were grown in culture in McCoy's medium with 10% FBS (Sigma-Aldrich) and 1% GlutaMAX (Gibco). HEK293T and BT16 cells were grown in culture in DMEM with 10% FBS and 1% GlutaMAX. BIN67, Kelly, and TTC549 cells were grown in culture in RPMI medium with 10% FBS and 1% GlutaMAX. CHLA-266 cells were grown in culture in IMDM with 20% FBS, 1% GlutaMAX, and 1% ITS (insulin, transferrin, selenium) (Gibco). Cell lines with Tet-inducible constructs were grown in culture in tetracycline-free FBS (Biowest) and induced by using 1 μg/mL doxycycline (Clontech) for the indicated time. MG132 (Sigma M7449) was added to cells at a concentration of 10 μM for 6 h, and MLN-4924 (CST 85923S) was added at a concentration of 1 μM for 24 h.

### Vectors and stable cell line generation
Lentiviral vectors were produced and titrated by the St. Jude Vector Production and Development Core as reported previously[16]. PHIP was knocked down in cell lines by using lentiviral transduction of CMV-driven shRNAs targeting PHIP (Mission TRCN0000127738; GCTGGAAGACAGTCTTTACTA) or scramble shRNAs (Addgene, #162001) in the presence of Polybrene (8 μg/mL, Santa Cruz Biotechnology) for 24 h, followed by selection using puromycin (1 μg/mL, Thermo Fisher Scientific) for 72 h. A multiplicity of infection (MOI) of 10 was used for G401, 293 T, and A204 cells; an MOI of 7 was used for TTC549 and BT16 cells; an MOI of 4 was used for BIN67 cells; and an MOI of 1 was used for Kelly cells.

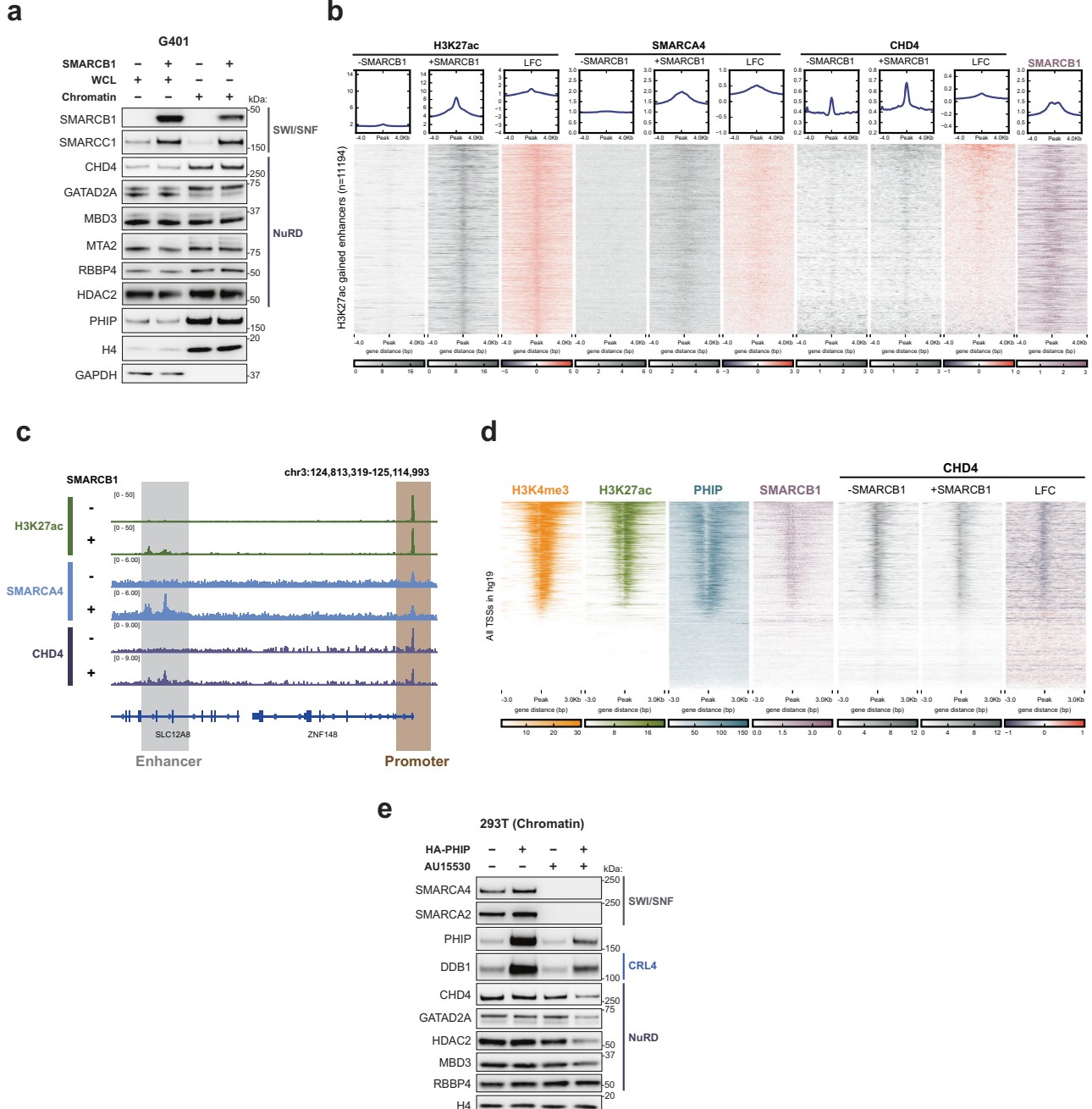

**Fig. 5 | SWI/SNF impairment stimulates suppression of NuRD by PHIP.**
**a** Evaluation of the rescue of SMARCB1 expression upon an abundance of NuRD subunits on chromatin in G401 RT cells. Western blots of whole cell lysate (WCL, lanes 1 and 2) and chromatin lysate (lanes 3 and 4) before and after restoration of SMARCB1 expression. GAPDH: WCL loading control, H4: chromatin loading control. Blot is representative of $n = 2$ biological replicates. **b** Evaluation of NuRD subunit CHD4 binding at enhancers that gain acetylation of H3K27 after SMARCB1 re-expression in G401 RT cells. Heatmap showing ChIP-seq coverage for H3K27ac ($n = 2$), SMARCA4 ($n = 2$), CHD4 ($n = 3$), and SMARCB1 ($n = 2$, purple) after SMARCB1 addback in G401 cells at enhancers that gain H3K27ac ($n = 11194$). The log2FC of signal +/− SMARCB1 is shown for each target (LFC, navy/red). Sorted by CHD4 LFC. **c** Genome track illustrating the effect of SMARCB1 re-expression upon CHD4 distribution at promoters, as compared to enhancers, in G401 RT cells. NuRD

(CHD4, purple), H3K27ac (green), and SMARCA4 binding (blue) are shown before and after SMARCB1 addback. **d** Evaluation of NuRD subunit CHD4 binding at promoters after SMARCB1 re-expression in G401 RT cells. Heatmap showing ChIP-seq coverage for H3K4me3 (orange, $n = 3$), H3K27ac (green, $n = 3$), PHIP (blue, $n = 3$), SMARCB1 (purple, $n = 2$), CHD4 in G401 with or without SMARCB1 (gray, $n = 3$), and the log2FC in CHD4 signal +/− SMARCB1 (LFC, navy/red) in G401 RT cells at all annotated TSSs in hg19. Sorted by H3K4me3 signal strength. **e** Evaluation of the effect of treatment with SMARCA2/4 degraders upon the activity of PHIP in targeting NuRD. Western blot of chromatin lysate from 293 T cells +/− PHIP overexpression +/− treatment with a SMARCA2/4 PROTAC (AU15530, 3 μM, 72 h) or vehicle control (DMSO). H4: chromatin loading control. Blot is representative of $n = 2$ biological replicates. Source data are provided as a Source Data file.

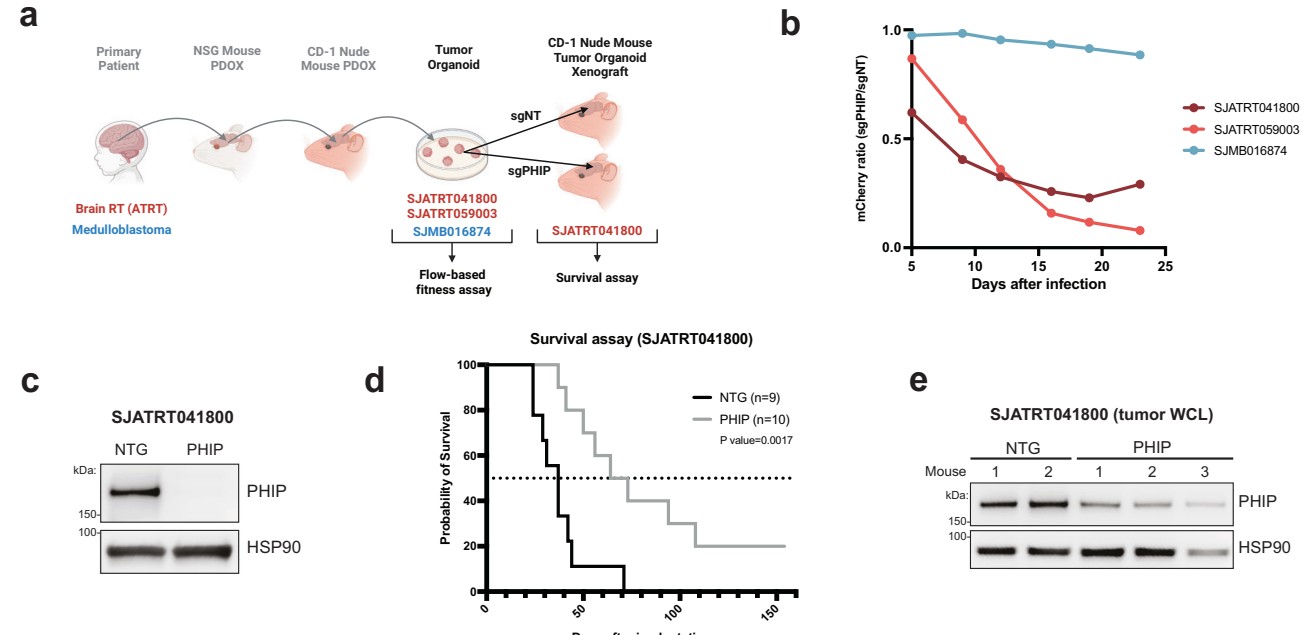

**Fig. 6 | Patient-derived in vivo models of RTs specifically depend upon PHIP.**
**a** Schematic outlining how the patient-derived tumor organoid lines and tumor organoid xenograft models were developed. Created in BioRender. Malone, H. (2026) https://BioRender.com/muu6g36. **b** Flow cytometry–based fitness assay comparing the effects of PHIP loss in two patient-derived AT/RT tumoroid lines that have inactivating mutations in *SMARCB1* (SJATRT041800 and SJATRT059003, red) and in a patient-derived medulloblastoma tumor organoid with intact SMARCB1 (SJMB016874, blue). Tumor organoids received control non-targeting (NT) mCherry-Cas9-sgNT or mCherry-Cas9-sgPHIP vectors. Flow cytometry analysis was performed longitudinally to measure the ratio of mCherry-positive cells. The plot shows the ratio of mCherry+ cells in sgPHIP conditions relative to sgNT at each time point. **c** Western blot of PHIP abundance in sgNT- or sgPHIP-treated pools of patient-derived AT/RT tumor organoid (SJATRT041800) on the day of orthotopic xenograft implantation into mice. HSP90: loading control. Blot is representative of *n* = 3 technical replicates. **d** Kaplan–Meier survival curves for mice orthotopically xenografted with either non-targeting sgNT (*n* = 9) or sgPHIP (*n* = 10) AT/RT tumor organoids (SJATRT041800). Data were analyzed with a log-rank (Mantel–Cox) test. *P* = 0.0017 for sgNT versus sgPHIP. The horizontal line indicates a 50% probability of survival. **e** Western blot of PHIP abundance in sgNT (*n* = 2 biological replicates) or sgPHIP (*n* = 3 biological replicates) tumors collected from patient-derived AT/RT tumor organoid (SJATRT041800) xenograft models. HSP90: loading control. Source data are provided as a Source Data file.

To knock out PHIP, complementary DNA oligonucleotides encoding sgRNAs for PHIP (sgPHIP1: AGGAAGTTACTCCCATCTGA, sgPHIP2: ATATGGCTACTCGTCCAGCA) or non-targeting (NTG) controls (sgNT3: TAGACGTCGTGAGCTTCAC) with appropriate overhangs compatible with BsmBI (Esp3I) digestion were purchased from IDT, and oligos were annealed. The lentiCRISPRv2-puro backbone (Addgene plasmid # 49535, a gift from Feng Zhang) was digested with *BsmBI*, and annealed oligonucleotides were ligated into the digested vector using T4 DNA ligase (New England Biolabs, M0202). Ligation reactions were transformed into competent E. coli, and bacterial colonies were selected on ampicillin-containing agar plates. Plasmid DNA was isolated, and sequences were confirmed by Plasmidsaurus sequencing. Lentiviral vectors were again produced and titrated by the St. Jude Vector Production and Development Core, and cells were transfected using the conditions described above. Cells were selected with puromycin for 7 days and then used for the indicated assays.

Generation of the isogenic GFP-inducible and SMARCB1-inducible 293T[SMARCB1 KO] and G401 lines was described previously[26]. Expression of GFP or SMARCB1 was induced by using 1 µg/mL doxycycline (Clontech) for 24 h. Flag-PHIP HEK293 cells were a gift from Dr. Ali Shilatifard[17].

To generate inducible HA-PHIP–expressing cell lines, 2X-HA-Strep-PHIP was inserted into the XLone-GFP PiggyBac expression vector (Addgene, #96930) between *Kpn*I and *Spe*I sites by using NEBuilder HiFi DNA Assembly (New England Biolabs, E2621S). To generate the ΔH-PHIP mutant, amino acids 1–161 of PHIP were deleted from the Xlone-PHIP construct by using a Q5 Site-Directed Mutagenesis Kit (New England Biolabs E0554S). Cells received the Super PiggyBac Transposase expression vector (System Biosciences, PB210PA-1) and

XLone-GFP, XLone-HA-PHIP, or XLone-HA-ΔH-PHIP via nucleofection, using the Lonza 4D-Nucleofector X unit (program EH-100, 100-µL cuvette). Stably expressing cells were selected using 2 µM blasticidin (Thermo Fisher). Expression of GFP, HA-PHIP, or ΔH-PHIP was induced with 1 µg/mL doxycycline (Clontech) for the indicated time. XLone-GFP (Addgene, plasmid #96930)[55] was a gift from Xiaojun Lian.

## Cell viability and proliferation assays
Cell growth was measured as described previously[51]. Cell lines were transduced with lentivirus with shRNAs or sgRNAs targeting PHIP and with non-targeting controls for 24 h then selected using 1 µg/mL puromycin for 7 days. Cells were then seeded into 96-well dishes with 8–16 replicate wells per condition. Wells were imaged, and confluence measurements were calculated every 8–12 h by using the Incucyte Live-Cell Analysis System (Essen BioScience). Confluency measurements were plotted using GraphPad Prism.

## PHIP CRISPR-Cas9 indel fitness assay (CelFi)
Indel fitness assays were performed by the Center for Advanced Genome Editing at St. Jude Children's Research Hospital as reported previously[28]. Precomplexed ribonuclear protein complexes containing 150 pmol of the indicated gRNAs (Synthego) (CAGE1359.PHIP.g5: CUGCUUCACCAGAACGUCUC, CAGE1359.PHIP.g6: CUUUGAAU-CAUGCUGUCCAG) and 50 pmol of *Sp*Cas9 protein (St. Jude Protein Production Facility) were transiently transfected into 1 million cells by nucleofection, using a Lonza 4D-Nucleofector X Unit (solution P3, program EH-100) in accordance with the manufacturer's instructions. Genomic DNA was collected from cells at days 3, 7, 14, 21, and 28 after nucleofection and used for targeted deep sequencing using gene-

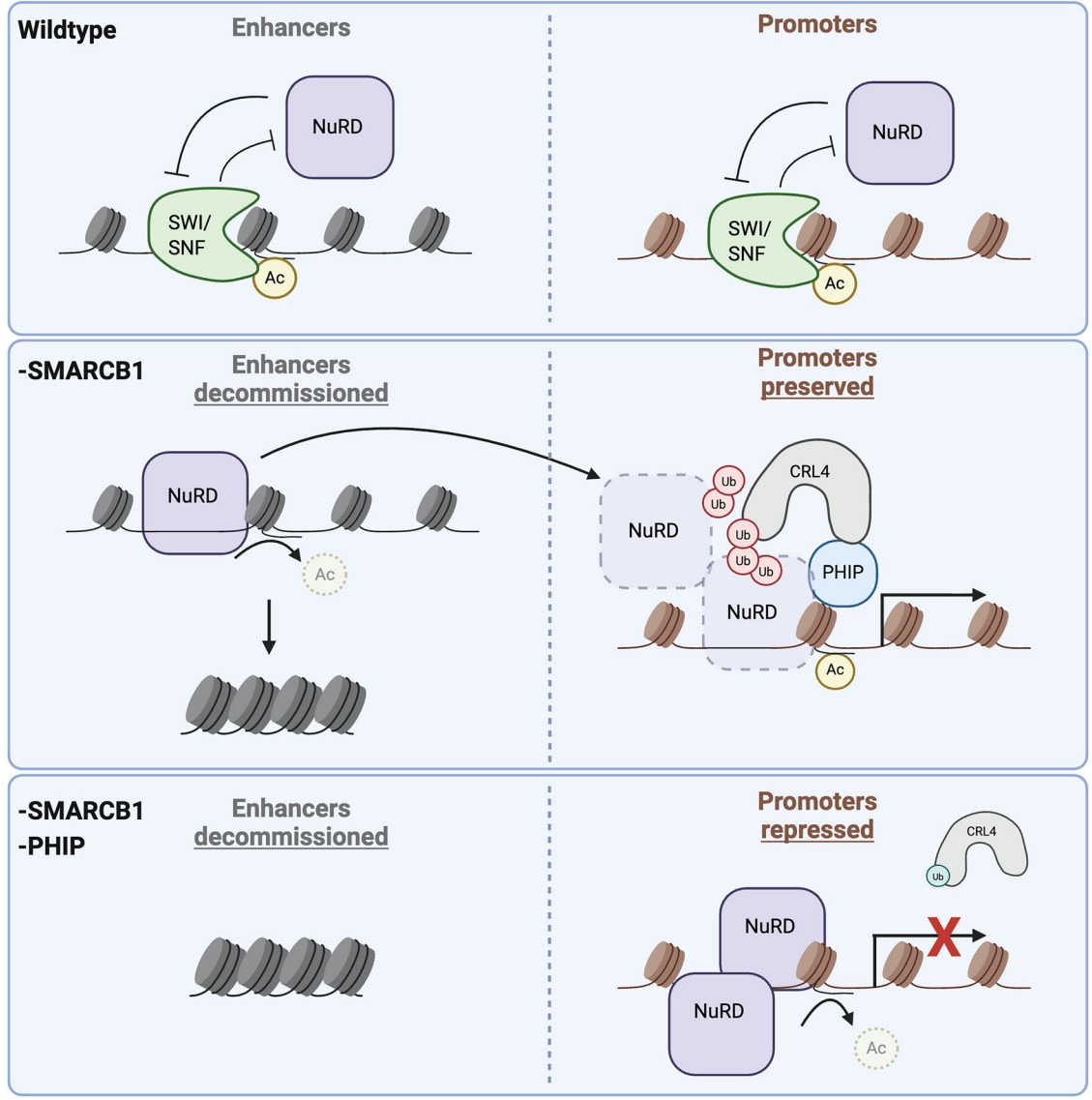

**Fig. 7 | Model.** In wildtype conditions, SWI/SNF and NuRD counterbalance one another at enhancers and promoters to control gene expression. When SWI/SNF is impaired (by inactivating mutations in *SMARCB1*, by the loss of SMARCA2 and SMARCA4 ATPases, or by treatment with a SMARCA2/4 PROTAC), enhancers are deacetylated and silenced and NuRD complexes accumulate at promoters. PHIP-mediated ubiquitination antagonizes NuRD to preserve promoter activation and gene expression. In SMARCB1-deficient cells, inactivation of PHIP causes NuRD complexes to accumulate at promoters where they deacetylate histones and silence gene expression. Created in BioRender. Malone, H. (2026) https://BioRender.com/kv89m1u.

specific primers with partial Illumina adapter overhangs. CRIS.py[56] was used for analysis, and out-of-frame, in-frame, and 0-bp indels were quantified and plotted using Prism.

### Analysis of raw data from DepMap
Avana log-fold changes values were downloaded from DepMap (v25Q3[57]) and subset to PHIP guides (sgPHIP1: AGGAAGTTACTCCCATCTGA, sgPHIP2: ATATGGCTACTCGTCCAGCA, sgPHIP3: GGTAGCTTGGAGTCGAAGGG, sgPHIP4: TCTGGTAGCTTGGAGTCGAA). Average logFC was computed for each guide. Average logFC values were stratified into two groups based on lineage MRT/ATRT/SCCOHT and "other". Violin plots were generated in R (v4.3.3) by using ggplot2 (v4.0.0[58]), and a *t*-test (ggpubr v0.6.0[59]) was performed on the means of the two groups.

### siRNA knockdown
Cells were treated with non-targeting siRNAs or with individual siRNAs against the indicated targets. Cells were reverse-transfected with siRNAs purchased from Horizon Discovery (formerly Dharmacon) (see Supplemental Table 1) for 6 h. Master mixes containing siRNAs diluted in Opti-MEM I Reduced Serum Medium (Thermo Fisher) were combined at a 1:1 ratio with Lipofectamine RNAiMAX Transfection Reagent (Thermo Fisher), diluted in Opti-MEM for a final siRNA concentration of 20 nM, and incubated for 15 min at room temperature. Cells were trypsinized and resuspended in Opti-MEM I Reduced Serum Medium then plated with the siRNA master mix. After 6 h, the medium was replaced by the appropriate growth medium for the cell type and incubation was continued for 48 h.

### Protein extraction
**Whole-cell extract.** Cells were lysed in RIPA lysis buffer (Thermo Fisher) supplemented with protease inhibitors (Halt Protease Inhibitor Cocktail; Thermo Fisher) and nuclease (Pierce Universal Nuclease; Thermo Fisher) for 1 h on ice with frequent vortex mixing. The mixture was then centrifuged at $16,000 \times g$ for 10 min. Lysates were quantified using a BCA assay (Thermo Fisher).

**Nuclear extract.** Cells were resuspended in hypotonic Buffer A (20 mM HEPES pH 7.4, 10 mM KCl, 0.2 mM EDTA, with protease inhibitors) and allowed to swell on ice for 10 min. The membranes were then lysed by adding NP-40 to 0.6%. The nuclei were concentrated by centrifugation for 1 min at 16,000 × g, and the cytoplasmic lysate was discarded. The nuclei were resuspended in IP buffer with nuclease (20 mM phospho-buffer pH 7.5, 150 mM NaCl, 10% glycerol, 0.2% Triton X-100, protease inhibitors, 2.5 μL/mL Pierce Universal Nuclease [Thermo Fisher]), placed on a rotary mixer for 20 min at 4 °C, then incubated at room temperature for 30 min with frequent pipetting. Lysates were then cleared by centrifugation at 16,000 × g for 10 min, after which the nuclear extract was quantified using a BCA assay.

**Subcellular fractionation.** Cells (15 million) were resuspended in 1.5 mL of hypotonic Buffer A (20 mM HEPES pH 7.4, 10 mM KCl, 0.2 mM EDTA, protease inhibitors) and allowed to swell on ice for 10 min. The membranes were then lysed by adding 90 μL of 10% NP-40 (0.6%). The nuclei were collected by centrifugation for 1 min at 16,000 × g, and the cytoplasmic fraction was saved. The nuclei were resuspended in 600 μL of IP buffer (20 mM phospho-buffer pH 7.5, 150 mM NaCl, 10% glycerol, 0.2% Triton X-100, protease inhibitors) and placed on a rotary mixer for 20 min at 4 °C. Insoluble chromatin was collected by centrifugation at 16,000 × g for 10 min, and the nuclear soluble fraction was retained. Chromatin pellets were then resuspended in 250 μL of chromatin extraction buffer (20 mM phospho-buffer pH 7.5, 150 mM NaCl, 10% glycerol, 0.2% Triton X-100, protease inhibitors, 20 μL/mL Pierce Universal Nuclease [Thermo Fisher]) and incubated at room temperature for 30 min with frequent pipetting. Lysates were centrifuged at 500 × g for 3 min, and chromatin lysate was retained. The cytoplasmic, nuclear soluble, and chromatin fractions were quantified using a BCA assay.

**Histone extraction for immunoblotting.** Histones were extracted from cells by using the EpiQuik Total Histone Extraction Kit (Epigen-Tek) supplemented with phosphatase and protease inhibitors. Histone lysates were quantified using a BCA assay.

## Western blot analysis

Protein from cell lysates was subjected to electrophoresis and transferred overnight at 4 °C onto a 0.45-μm nitrocellulose membrane (Bio-Rad, 162-0115). Membranes were blocked with 5% BSA in Tris-buffered saline with 0.1% Tween 20 (TBST) and incubated with primary antibodies (see Supplemental Data 1). Membranes were washed three times with TBST then incubated with peroxidase-conjugated secondary antibodies for 1 h at room temperature. After incubation with secondary antibody, the membranes were washed three times with TBST and imaged using a Bio-Rad ChemiDoc or Licor Odyssey XF imaging system. The images obtained were analyzed with ImageLab and ImageStudio software, respectively.

## Immunoprecipitation of nuclear proteins

Nuclear lysate was quantified by BCA assay. Samples were diluted to a final concentration of 1 mg/mL and precleared with washed Dynabeads Protein G (Thermo Fisher Scientific, 10003D) for 1.5 h at 4 °C. Inputs (5%) were saved, and the nuclear lysate was incubated overnight at 4 °C with appropriate antibodies (see Supplemental Data 1). Samples were then incubated with Dynabeads Protein G in IP buffer for 1 h at 4 °C. Bead–antibody–protein conjugates were washed three times in cold IP buffer for 3 min at 4 °C. The samples were then eluted with 2× LDS at 95 °C for 10 min. For the subsequent Western blot analyses, the previously described protocol was followed, using light chain–specific secondary antibodies.

## Glycerol gradient fractionation

Glycerol gradient fractionation was performed as described previously[16]. Nuclear extracts were prepared from siCtrl- and siPHIP-

treated cells as described above. For each condition, 1 mg of nuclear protein, quantified by BCA, was layered onto a 10–30% glycerol gradient prepared in a 14 × 95 mm ultracentrifuge tubes (Beckman Coulter, Brea, CA; Cat. #344060) using the Biocomp Gradient Station (Fredericton, NB, Canada, Model 153). Gradients were subjected to ultracentrifugation at 200,000 × g for 16 h at 4 °C using an SW40 rotor. 20 fractions were collected per sample by using the Biocomp Gradient Station (Model 153) piston fractionator. Proteins from each fraction were subsequently analyzed by immunoblotting to determine the distribution of CRL4 and NuRD subunits.

## In vitro ubiquitination assays

FLAG-tagged PHIP-CRL4 was prepared from HEK293 cells as described previously[17]. In vitro ubiquitination reactions were performed in a final volume of 50 μL, using 100 nm E1 (recombinant human ubiquitin-activating enzyme [UBE1]) (Bio-Techne R&D Systems, #E2-305), 500 nM E2 (recombinant human UbcH5a/UBE2D1) (Bio-Techne R&D Systems, #E2-616), 10 μg of ubiquitin (Bio-Techne R&D Systems, U-100H), 60 mM ATP (Thermo Fisher Scientific, R0441), and 5 μL of FLAG-PHIP eluate in reaction buffer (final concentration: 50 mM Tris pH 7.5, 5 mM MgCl₂, 0.1% Tween-20, 0.5 mM DTT). Recombinant FLAG-CHD4 (Active Motif, 81385) or 6xHIS-SUMO-HDAC1 (MedChemExpress, HY-P72262) were added to the indicated assays. Reaction mixtures were incubated for 2 h at 30 °C then the reactions were terminated by adding 15 μL of 4× SDS sample buffer and incubating for 10 min at 95 °C. Reaction products were subjected to Western blot analysis.

## Histone extraction and LC-MS/MS analysis

Histones were extracted for PTM analysis by using a protocol described previously[60]. Cell pellets were washed twice with nuclear isolation buffer (15 mM Tris-HCl, pH 7.5, 60 mM KCl, 15 mM NaCl, 5 mM MgCl₂, 1 mM CaCl₂, 250 mM sucrose, 0.6 mM AEBSF, 1 mM DTT, 0.1 μM microcystin, 10 mM sodium butyrate) and concentrated by centrifugation at 700 × g for 5 min at 4 °C. Cells were then lysed in nuclear isolation buffer + 0.3% NP-40 for 5 min on ice, then chromatin was concentrated by centrifugation at 700 × g for 5 min. Chromatin pellets were resuspended in 5 volumes of H₂SO₄ and agitated for 2 h at 4 °C to solubilize histones. Lysates were centrifuged at 3400 × g for 5 min, then histones were precipitated from the supernatant by adding 1/3 volume of 100% TCA and incubating the mixture at 4 °C overnight. Histones were concentrated by centrifugation at 3400 × g and subsequently washed twice with cold acetone + 0.1% HCl. Pellets were dried at room temperature then finally resuspended in water and quantified using a DC protein assay (Bio-Rad).

The histones were extracted and prepared for chemical derivatization and digestion as described previously[61,62]. In brief, the lysine residues from histones were derivatized with the propionylation reagent (1:2 reagent:sample ratio) containing acetonitrile and propionic anhydride (3:1), and the solution pH was adjusted to 8.0 with ammonium hydroxide. The propionylation was performed twice, and the samples were dried on a SpeedVac vacuum concentrator. The derivatized histones were then digested with trypsin at a 1:50 ratio (wt/wt) in 50 mM ammonium bicarbonate buffer at 37 °C overnight. The N-termini of histone peptides were derivatized with the propionylation reagent twice and dried on the SpeedVac. The peptides were desalted with self-packed C18 stage tips. The purified peptides were then dried and reconstituted in 0.1% formic acid. A liquid chromatography–tandem mass spectrometry (LC-MS/MS) system consisting of a Vanquish Neo UHPLC System coupled to an Orbitrap Exploris 240 or Ascend mass spectrometer (Thermo Fisher Scientific) was used for peptide analysis. Histone peptide samples were maintained at 7 °C on a sample tray in LC. Peptides were separated on an Easy-Spray™ PepMap™ Neo nano-column (2 μm, C18, 75 μm × 150 mm) at room temperature with a mobile phase. The chromatography

conditions consisted of a linear gradient from 2 to 32% solvent B (0.1% formic acid in 100% acetonitrile) in solvent A (0.1% formic acid in water) over 48 min and then 42% to 98% solvent B over 12 min at a flow rate of 300 nL/min. The mass spectrometer was programmed for data-independent acquisition (DIA). One acquisition cycle consisted of a full MS scan and 35 DIA MS/MS scans of 24 m/z isolation width, starting from 295 to 1100 m/z. Typically, full MS scans were acquired in the Orbitrap mass analyzer across 290–1200 m/z at a resolution of 70,000 or 120,000 in positive profile mode, with an injection time of 50 ms and an AGC target of 1e6 or 200%. MS/MS data from higher-energy collisional dissociation (HCD) fragmentation was collected in the Orbitrap. These scans typically used a normalized collision energy (NCE) of 30 or 25, an automatic gain control (AGC) target of 1000%, and a maximum injection time of 60 ms. Histone MS data were analyzed with EpiProfile (version 2.0)[63].

## IP-MS

Immunoprecipitated samples were washed three times with IP buffer and processed for MS analysis by on-bead digestion. Before trypsin digestion, the beads were resuspended in 50 mM $NH_4HCO_3$, then reduction with DTT, alkylation with iodoacetamide, and digestion with trypsin were performed overnight at 37 °C. The peptides in solution were transferred to a new tube, and the beads were washed twice with 50% acetonitrile to collect the remaining peptides. The resulting peptides were dried using a SpeedVac vacuum concentrator. The peptides were desalted with self-packed C18 stage tips. The purified peptides were then dried and reconstituted in 0.1% formic acid. An LC-MS/MS system consisting of a Vanquish Neo UHPLC System coupled to an Orbitrap Exploris 240 or Ascend mass spectrometer (Thermo Scientific) was used for peptide analysis. Peptides were separated on an Easy-Spray™ PepMap™ Neo nano-column (2 μm, C18, 75 μm × 150 mm) at room temperature with a mobile phase. The chromatography conditions consisted of a linear gradient from 2 to 32% solvent B (0.1% formic acid in 100% acetonitrile) in solvent A (0.1% formic acid in water) over 75 min and then 45% to 98% solvent B over 15 min at a flow rate of 300 nL/min. The mass spectrometer was programmed for data-dependent acquisition (DDA) mode. The MS1 was acquired over a mass range of 295–1100 m/z with a resolution of 120,000, an AGC target of 300%, and a maximum injection time of 50 ms. For the MS2, the top 20 most intense ions were selected for MS/MS by HCD at an NCE of 30, with a resolution of 15,000, an AGC target of 75%, and a maximum injection time of 60 ms. Raw files were processed using FragPipe (version 19.1)[64]. Reviewed human protein sequences were downloaded from UniProt (UP000005640). Carbamidomethyl (C) was set as a static modification, and oxidation (M) and acetylation (protein N-terminus) were selected as variable modifications. Search parameters included tryptic digestion (C-terminal to KR) with 2 missed cleavages, precursor and fragment mass tolerances of 20 ppm, peptide and protein-level FDR cutoffs of 1%, peptide length 7–50, and peptide mass range 500–5000 Da.

## RNA-seq

RNA-seq was performed as described previously[16]. RNA was extracted from 1 million cells with Trizol, then purified using a DirectZol Miniprep Plus Kit (Zymo Research). RNA was quantified using the Quant-iT RiboGreen Assay (Thermo Fisher Scientific), and its quality was validated with the Agilent 4200 TapeStation RNA ScreenTape assay system. Libraries were generated with the Illumina Stranded Total RNA Prep (Illumina, 20040525) and quantified with a Quant-iT PicoGreen dsDNA Assay Kit (Thermo Fisher Scientific). Library quality control was performed using the Agilent 4200 TapeStation D1000 ScreenTape assay. Next-generation sequencing was performed at the Hartwell Genome Sequencing Facility at St. Jude Children's Research Hospital, using the Illumina NovaSeq 6000 Sequencing System (50-bp, paired-end reads).

## ChIP-seq

ChIP-seq was performed as described previously[65]. Twenty million cells per replicate were washed in PBS, resuspended in Fixing Buffer A (Covaris truChIP Chromatin Shearing Kit), fixed in 1.1% formaldehyde for 10–12 min with rotation, then quenched with 0.125 M glycine for 5 min with rotation. Cells were pelleted and then washed with PBS, and 10% (2 million) fixed *Drosophila* S2 cells were added. Cell pellets were resuspended in Lysis Buffer B and incubated for 10 min at 4 °C to isolate the nuclei. The nuclei were concentrated by centrifugation, resuspended in Wash Buffer C, and incubated for 10 min at 4 °C. Washed nuclei were resuspended in Shearing Buffer D3, transferred to Covaris shearing tubes (Covaris milliTUBE 1 ml AFA Fibre), and sheared using a Covaris E220 sonicator (peak incident power = 140 W, cycles per burst = 200, duty factor = 5%, water level [run] = 2, time = 12 min for G401 cells). Efficient shearing of the chromatin was confirmed by gel electrophoresis.

Each immunoprecipitation used chromatin from 5 million to 20 million cells. Spike-in *Drosophila* antibodies (Active Motif, 61751) were used according to the manufacturer's recommendations. Pre-cleared chromatin was incubated with antibodies overnight at 4 °C with rotation (see Supplemental Data 1). Antibody-bound chromatin was then isolated using Dynabeads Protein G, washed, and eluted in elution buffer (1% SDS, 0.1 M NaHCO$_3$ in RNase- and DNase-free water) for 30 min at 65 °C with shaking. Eluted DNA–protein complexes were de-crosslinked overnight and purified using a MiniElute PCR Purification Kit (Qiagen). Libraries were prepared using a KAPA HyperPrep Kit (KK8504) and validated using the Agilent TapeStation. Next-generation sequencing was performed at the Hartwell Genome Sequencing Facility at St. Jude Children's Research Hospital, using the Illumina NovaSeq 6000 Sequencing System (50-bp, paired-end reads).

## Propidium cell cycle analysis

One million cells per condition and replicate were trypsinized and dissociated into a single-cell suspension. Cells were washed once with PBS then resuspended in 1 mL of propidium iodide (PI) solution (0.05 mg/mL propidium iodide; 0.1% [w/v] sodium citrate; 0.1% [v/v] Triton X-100), vortex mixed, and incubated for 30 min at room temperature in the dark. Samples were subsequently treated with 10 μL of 0.2 mg/mL RNase (Thermo Fisher, #EN0531) for 30 min at room temperature. Samples were filtered through a 40-μM mesh strainer then the fluorescent signal was acquired by flow cytometry on a BD FAC-Symphony A5 SE Cell Analyzer (BD Biosciences) controlled by DiVa software version 9.6. The fluorescence was analyzed using ModFit software (Verity Software House).

## Cell culture (tumor organoids)

AT/RT and medulloblastoma patient-derived tumor organoids were provided by Martine Roussel (St. Jude). These were maintained at 37 °C with 5% CO$_2$ in tumor stem medium (TSM) full medium as described previously[45]. The medium consists of a 50:50 mix of DMEM:F12 (ThermoFisher) and Neurobasal medium A (Thermo-Fisher) supplemented with B27 without vitamin A (Gibco), N2 (Gibco), Heparin (5IU/mL, ThermoFisher), EGF (31.25 ng/uL, Pepro-tech), FGF2 (31.25 ng/uL, Peprotech), platelet-derived growth factor AA (PDGF-AA; 10 ng/mL, Peprotech), and PDGF-BB (10 ng/mL, Peprotech). Fresh media was added every 3-4 days, and cells were passaged every 7–10 days at a 1:3–1:6 ratio. To passage, cells were mechanically dissociated, washed with 5-10 mL of cold DMEM, centrifuged at 300 x G for 5 min at 4 °C, and replated in fresh TSM. All tumoroid models are stored in the biobank of the Princess Máxima Center or St. Jude Children's Research Hospital and are available to the scientific community according to the rules and regulations under which the patients and parents gave informed consent for donating the tissue.

## Competition proliferation assay

sgRNAs targeting PHIP (sgPHIP1: AGGAAGUUACUCCCAUCUGA, sgPHIP2: AUAUGGCUACUCGUCCAGCA) or non-targeting (NTG) controls (sgAAVS1: ACCCCACAGUGGGGCCACUA, sgLacZ: UGCGAAUACGCCCACGCGAUGGG) were cloned into the LentiCRISPRv2-mCherry backbone by the Center for Advanced Genome Engineering (CAGE) at St. Jude. LentiCRISPRv2-mCherry (Addgene plasmid #99154) was a gift from Agata Smogorzewska. Proper sgRNA insertion and plasmid validation were confirmed by Plasmidsaurus long-read sequencing. Tumor organoid cultures were passaged as described previously[45], and 1 million cells were seeded in ultra-low attachment T25 flasks (Corning, #4616) for each condition (NTG or sgPHIP). Tumor organoids were infected three times daily at a multiplicity of infection (MOI) of 5 infectious units (IU) per cell, with 3 h between infections, for 2 days (for a total of six infections), with the inoculum being supplemented with 4 µg/mL polybrene (Sigma-Aldrich, #TR-1003-G). The morning after the last infection, cells were concentrated by centrifugation and resuspended in 5 mL of fresh TSM full medium. Forty-eight hours after the final infection, an aliquot of cells was harvested from the culture for each condition, dissociated into a single-cell suspension, and subjected to flow cytometry analysis using an LSRFortessa instrument (BD Bioscience) to measure the baseline/initial mCherry fluorescence. Flow cytometry analysis was performed over time, and cell pellets were collected to measure the editing efficiency over time. Genomic DNA was collected from cell pellets and used for targeted deep sequencing, using gene-specific primers with partial Illumina adapter overhangs. CRIS.py[56] was used for analysis, and out-of-frame indels were quantified and plotted relative to day 0 by using Prism.

## In vivo survival analysis

sgRNAs targeting PHIP or non-targeting controls (see above) were cloned into the LentiCRISPR v2 backbone by the St. Jude CAGE. LentiCRISPR v2 (Addgene plasmid #52961) was a gift from Feng Zhang. Proper sgRNA insertion and plasmid validation were confirmed by Plasmidsaurus sequencing. SJATRT041800 tumor organoid cells were dissociated, and 5 million cells per condition were seeded in ultra-low-attachment T75 flasks (Corning, #3814). Cells were infected three times daily at an MOI of 5 IU per cell, with 3 h between infections, for 2 days (for a total of six infections), with the inoculum being supplemented with 4 µg/mL polybrene (Sigma-Aldrich, #TR-1003-G). The morning after the final infection, tumor organoid cells were concentrated by centrifugation, resuspended in 15 mL of fresh TSM full medium, and replated in ultra-low-attachment T75 flasks. Seventy-two hours after the final infection, the tumor organoid cells were dissociated and replated in 20 mL of fresh TSM medium supplemented with 2 µg/mL puromycin (Thermo Fisher, #A1113803) in ultra-low-attachment T75 flasks. Puromycin selection lasted for 72 h, after which the tumor organoid cells were dissociated and counted.

All animal studies were approved by the St. Jude Children's Research Hospital Animal Care and Use Committee and were performed in accordance with best practices outlined by the NIH Office of Laboratory Animal Welfare. Animals were housed under a 12 h–12 h light–dark cycle (light on at 06:00 and off at 18:00) at 68–70 °F (20–22 °C) with humidity levels maintained at 30–70%. Food and water were provided ad libitum. Power analysis was conducted to determine the sample size. Experiments were not randomized, and the investigators were not blinded to allocation during endpoint monitoring.

Tumor organoid cells (200,000/mouse) were orthotopically implanted in the cortices of naïve female CD1 nude mice (Charles River Laboratories, #086NU/NUCD1) aged approximately 8–12 weeks. Implantations were performed as described previously[44]. Mice were anesthetized by intraperitoneal administration of a ketamine/xylazine cocktail sufficient to achieve and maintain a surgical depth of anesthesia for approximately

30–40 min. Once anesthetized, animals were positioned in a stereotaxic frame and immobilized. The scalp was incised slightly lateral to the midline, after which the underlying fascia was carefully cleared and the skin retracted using a sterile cotton-tipped applicator to expose the skull surface. A dental drill was used to create a small square craniotomy in the skull, which was subsequently lifted away using Dumont forceps. The exposed area was rinsed with sterile saline, and residual fluid and bone debris were removed using gentle suction. Tumor cells suspended in Matrigel (5 µL total volume) were delivered into the right cerebral hemisphere using a blunt-tip Hamilton syringe under stereotaxic guidance. Following injection, animals were released from the stereotaxic apparatus, and the scalp incision was closed with two surgical wound clips. Mice were transferred to clean cages placed on a warming surface and monitored until recovery from anesthesia. Supplemental heat was provided for 48 h postoperatively. Wound clips were removed 7–10 days after surgery.

Tumor growth was monitored by MRI weekly, beginning 7 days post-implantation. Tumor-bearing mice were monitored for signs of CNS disease daily. The study endpoint was defined by neurologic symptoms (seizure, cranial swelling, ataxia). Mice reaching these endpoints were euthanized, and their tumors were collected for subsequent analysis. Survival curves were analyzed using the Cox regression or log-rank test.

Two million cells per condition were replated in culture for parallel in vitro monitoring. The remaining cells were concentrated by centrifugation and used for molecular analysis to validate the initial PHIP knockout by Western blot analysis and the editing efficiency by deep sequencing.

## NGS data processing

Sequence data was processed as reported previously[16]. Adapters were trimmed from raw reads in FASTQ files with Trim-Galore (v0.4.4, https://www.bioinformatics.babraham.ac.uk/projects/trim_galore/), using the cutadapt program[66]. QC was performed on trimmed files by using FastQC (https://www.bioinformatics.babraham.ac.uk/projects/fastqc/) with the quality score cutoff set to Q20. Paired-end ChIP-seq reads were mapped to a hybrid human–Drosophila reference genome (merged hg19/GRCh37.p13 and dm6 genomes) with bwa aln, followed by bwa samse (v0.7.12-r1039[67]) with the -K flag set to 10,000,000. The output was converted to BAM format with samtools (v1.2[68]), and reads mapped to the human reference genome were extracted for further analysis by using samtools. RNA-seq paired-end reads were mapped to the human reference genome (hg19/GRCh37.p13) with STAR (2.7.1a[69]). Duplicated reads were identified and marked using the bamsormadup tool from biobambam2 (v2.0.87[70]).

## ChIP-seq

**Mapping and peak calling.** Cross-correlation analysis was conducted with SPP (v1.11[71]), and uniquely mapped reads were extracted with samtools[68], extended with BEDtools (v2.24.0[72]), using the fragment size calculated by the cross-correlation analysis. UCSC tools (v4[73]) was used to convert bam files to bigWig track files. Peaks were called using MACS2 (v2.2.7.1[74]) with -nomodel -q 0.05 flags (high confidence peaks). Low-confidence narrow peaks were also called using a more relaxed criterion (-q 0.5 flag). Reproducible peaks of biological replicates were called using a previously reported approach[75]. Reproducible peaks were defined as those loci with overlapping high-confidence peaks in more than one replicate or as loci called as high-confidence peaks in one replicate and as high- or low-confidence peaks in all other replicates. The coordinates of the final reproducible peaks were based on the union of coordinates of the overlapping high-confidence peaks. For visualization on heatmaps, peaks that overlapped with annotated TSSs were identified using intersectBed.

**Normalization.** *Drosophila* reads were extracted from hybrid bam files by using samtools (v1.2[68]) and the dm6 chromosome names (v1.2). The number of uniquely aligned, non-duplicated dm6 reads was used to compute the % spike in reads over all reads sequenced for each sample and input. A ratio of dm6 reads was computed for each sample, compared to the dm6 reads generated for each input sample. Scaling factors were computed as the input scaled dm6 ratio for each sample, divided by the largest dm6 ratio for the mark. For generating bigWig files, a library size normalization was included, defined as 15 million / uniquely aligned, non-duplicate reads * the previously computed scaling factor. Code for spike-input normalization has been uploaded to GitHub: [https://github.com/jamyers2358/Malone_PHIP_SWISNF_Dependency].

**Differential analysis.** Combined count matrix files of reads within each reference peak were quantified using the intersect command from pybedtools (v0.8.1[72,76]) and were used as input for downstream differential testing in R (v4.3.3/v4.5.0). To perform statistical tests of differences among experimental groups, the trimmed mean of M-value scale factors was estimated using edgeR (v0.16) and a limma-voom (v3.58.1) empirical Bayes moderation to establish significant differences. Significant differential binding of targets was defined as significantly gained (log2FC > 0 and FDR < 0.05) or significantly lost (log2FC < 0 and FDR < 0.05). The scaling factors calculated above were applied during differential testing for input-spike-in–scaled experiments.

**Visualizing results of differential analysis by using heatmaps.** Hierarchically clustered and row z-score–scaled heatmaps were generated in R (v4.5.0) using pheatmap (v1.0.13[77]), based on significant differentially expressed genes or modified/bound regions.

A hierarchically clustered heatmap (pheatmap v1.0.13[77]) was generated for expression values (TPM) of genes downregulated upon the loss of PHIP (4115 log2FC < 0 and adjusted $P < 0.05$) across samples from the following conditions: control, PHIP-overexpression, PHIP-knockdown, and PHIP-rescue. k-means clustering (kmeans_k = 2) was then applied to the genes to obtain two groups ($c1 = 2034, c2 = 2081$). The heatmap was then regenerated based on cluster assignment.

**Visualization.** For visualization of ChIP-seq samples without spike normalization, the density of mapped reads was converted to bigWig format and normalized to 15 million non-duplicated mapped reads to normalize for library size. Input-spike–normalized samples were library-size normalized, defined as 15 million/uniquely aligned, non-duplicate reads, and multiplied by the previously computed scaling factor. The signal for the three biological replicates was averaged for display in the main figure panels. Differential bigWigs were generated with averaged bigWig files, using the log2FC function of bigwigCompare. All heatmaps and metaplots of normalized, averaged coverage were generated using deepTools (v2.5.3[78]), using reference point mode. Matrix files were generated with computeMatrix, heatmaps with plotHeatmap, and metaplots with plotProfile. The integrative genome viewer (IGV v 2.11.3[79]) was used to visualize coverage at specific loci.

Several previously published ChIP-seq datasets were used in visualizations, including H3K4me1 in G401 cells (GSE180487[51]), SMARCB1 after SMARCB1 addback in G401 cells (GSE210636[80]), SMARCA4 after SMARCB1 addback in G401 cells (GSE210636[80]), and H3K27ac after SMARCB1 addback in G401 cells (GSE178490[81]).

**Peak annotation.** Reproducible peaks were annotated using HOMER (v4.9.1). Peaks assigned to genes and within ±2 kb of their promoters were defined as "proximal peaks." Peaks more than 2 kb from annotated TSSs were defined as "distal peaks." Several publicly available PHIP ChIP-seq datasets were used (HCT116[18] [GSM5696448], 293T[17] [GSE101646], and v6.5 mESCs[17] [GSE101646]).

## RNA-seq

Gene-level counts were quantified using rsem-calculate-expression, using BAM files generated by STAR. A combined count matrix with all samples was generated and used as input for differential testing. To perform statistical tests of differences between experimental groups, the trimmed mean of M-value scale factors was estimated using edgeR and a limma-voom empirical Bayes moderation to establish significant differences. Significantly differentially expressed genes (DEGs) were defined as those with log2FC > 1 and FDR < 0.05 or log2FC < 1 and FDR < 0.05. Hierarchically clustered and z-score–centered heatmaps were generated using pheatmap ([https://cran.r-project.org/web/packages/pheatmap/index.html](https://cran.r-project.org/web/packages/pheatmap/index.html)). Gene Ontology enrichment analysis was performed with the indicated gene sets by using ShinyGO (v0.82, [http://bioinformatics.sdstate.edu/go/](http://bioinformatics.sdstate.edu/go/)).

### Data integration

**BETA.** BETA (v1.0.7[29]) was run using ChIP-seq binding peaks and significantly differentially expressed genes. The -d flag was set to 100,000 to include only peaks within 100 kb of a gene's TSS. The regulatory potential score was calculated using $Sg = \sum ki = 1e - (0.5 + 4\Delta i) Sg = \sum ik = 1e - (0.5 + 4\Delta i)$. All peaks (k) within 100 kb of the TSS were considered. The distance between a binding site and the TSS is $\Delta$, which is proportional to 100 kb. $P$ values were calculated by the Kolmogorov–Smirnov test to measure the significance of the upregulated genes group or the downregulated genes group relative to the static genes group.

**GSEA.** Pre-ranked gene set enrichment analysis (GSEA) was performed in python by using GSEApy[82] (v1.1.2, [https://gseapy.readthedocs.io/en/latest/introduction.html](https://gseapy.readthedocs.io/en/latest/introduction.html)). Gene log2FC was used to rank genes in the gene list. Several custom gene sets were defined using internal and external differential expression analyses.

### Reporting summary

Further information on research design is available in the Nature Portfolio Reporting Summary linked to this article.

## Data availability

The RNA and ChIP-Seq data generated in this study have been deposited in NCBI's Gene Expression Omnibus database under the GEO series accession GSE315219 and GSE299672. Previously published ChIP-seq and RNA-seq datasets used in this study can be accessed under the following GEO accession codes: GSE101646[17], GSE189235[18], GSE180487[51], GSE210636[80], GSE178490[81], and GSE215024[16]. Results from the CRISPR screen can be accessed at DepMap ([https://depmap.org/portal/](https://depmap.org/portal/)). The raw IP-MS and histone-MS data generated in this study have been deposited to ProteomeXchange with the identifier PXD069047 [[https://massive.ucsd.edu/ProteoSAFe/dataset.jsp?accession=MSV000099353](https://massive.ucsd.edu/ProteoSAFe/dataset.jsp?accession=MSV000099353)]. Plasmids obtained from Addgene are available under the indicated accession numbers, and plasmids generated in this study are available from the corresponding author upon request. Source data are provided with this paper.

## Code availability

Custom code to perform the input spike-in normalization can be found on GitHub ([https://github.com/jamyers2358/Malone_PHIP_SWISNF_Dependency](https://github.com/jamyers2358/Malone_PHIP_SWISNF_Dependency)).

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

## Acknowledgements

This work was supported by grants from the National Cancer Institute (NCI) (R01-CA-113794, R01-CA-273455, and R01-CA-172152 to C.W.M.R.; P01-CA-96832 to M.F.R.; and F31-CA-278355 to H.A.M.); the National Institutes of Health (NIH) (R01-HD-106051 and P01-CA-196539 to B.A.G.); CURE AT/RT Now (to C.W.M.R.); the Garrett B. Smith Foundation (to C.W.M.R.); the St Jude Children's Research Hospital Collaborative Research Consortium on Chromatin Regulation in Pediatric Cancer (to C.W.M.R. and B.A.G.); and the Pediatric Cancer Dependencies Accelerator of the Broad Institute, Dana-Farber Cancer Institute, and St. Jude Children's Research Hospital (peddep.org) (to C.W.M.R. and M.F.R.). H.A.M., J.D.F., and S.J.K. are supported by the St. Jude Graduate School of Biomedical Sciences. The authors thank members of the Roberts laboratory, Sandi Radko-Juettner, Robert Mobley, and Adam Durbin (St. Jude) for discussions; Keith A. Laycock, PhD, ELS, for scientific editing of the manuscript; Junmin Peng and Zhiping Wu (St. Jude) for help with proteomics experiments; Elizabeth Stewart and Asa Karlstrom (St. Jude) for assistance with xenograft experiments; Flore A. Cuisin (University Paris Cité) for assistance with patient-derived tumor organoid experiments; and the following St. Jude core facilities for their support: the Center for Applied Bioinformatics for sequencing and analysis, the St. Jude Vector Laboratory Shared Resource for lentiviral production, the Flow Cytometry and Cell Sorting Shared Resource for flow cytometry analysis and FACS sorting, the Animal Resources Center for animal care, the Center for In Vivo Imaging and Therapeutics for in vivo imaging, and the Center for Advanced Genome Editing for gRNA design, validation, and sequence analysis. The St. Jude Core facilities are supported by National Cancer Institute Cancer Center Support Grant (NCI-CCSG-2 P30-CA-021765) and by the American Lebanese Syrian Associated Charities (ALSAC) of St. Jude Children's Research Hospital. The content is solely the responsibility of the authors and does not necessarily represent the official views of the National Institutes of Health.

## Author contributions

H.A.M. and C.W.M.R. conceived the experiments and the study design. J.A.M., J.J.N., and H.A.M. performed all data processing, computational and statistical analyses, and data visualization. H.A.M., E.G.G., M.A.M., T.C.M., R.L.H., and S.J.K. contributed to cell line and biochemical experiments. H.A.M., E.G.G., and T.C.M. performed ChIP-Seq experiments. H.A.M., T.C.M., and R.L.H. performed RNA-seq experiments. J.D.F., S.R., and M.F.R. designed, supervised, and performed experiments in patient-derived tumor organoids and orthotopic xenograft AT/RT models. S.M.P-M. and B.S.H. designed and performed CRISPR-mediated fitness assays. B.A.G. and F.N.D.L.V. performed and analyzed histone PTM MS, and IP-MS experiments. H.A.M., J.A.M., E.G.G., M.A.M., J.D.F., T.C.M., J.J.N., S.R., R.L.H., S.J.K., F.N.D.L.V., B.S.H., S.M.P-M., B.A.G., M.F.R., J.F.P., and C.W.M.R. contributed to the interpretation of results. H.A.M. and C.W.M.R. wrote the manuscript with input from all co-authors.

## Competing interests

The authors declare no competing interests.
