## [Transparent Peer Review file · Nature Communications]

PHIP suppresses NuRD to enable the growth of SWI/SNF-mutant cancers

Corresponding Author: Dr Charles Roberts

Version 0:

Reviewer comments:

Reviewer #1

(Remarks to the Author)

The manuscript by Malone et al. entitled "PHIP suppresses NuRD to enable the growth of SWI/SNF-mutant cancers" the authors identified PHIP as a dependency for cancers with disrupted SWI/SNF function. SWI/SNF chromatin remodelers are large multi-subunit complexes mutated in up to 20% of cancers and thus the identification of a vulnerability in these SWI/SNF-mutated cancers is an important advance in the field and has therapeutic implications for the treatment of SWI/SNF-mutated cancers. The authors identified PHIP through a CRISPR/Cas9 screen that compared dependencies between cancer cells, Rhabdoid tumors (RT), and small cell carcinomas of the ovary, hypercalcemic type (SCCOHT). PHIP showed as a dependency on both RTs and SCCOHT, but not in all other cancer types. The authors validated this finding in 293T cells showing that inactivation of SMARCB1 and PHIP resulted in reduced growth, but not by inactivating these genes independently. Using RT cell lines, the authors downregulated PHIP and showed that the inactivation of this factor resulted in downregulation of gene expression of various pathways associated with cell proliferation, thus explaining why RT cells would require this factor for growth. They also showed that PHIP can be found on the promoters of the genes that are downregulated upon PHIP inactivation. The authors found PHIP in a complex with ubiquitin ligases and did IP-MS to identify substrates, which resulted in the identification of the NuRD complex as a target. The authors did a series of experiments using ChIP-Seq in RT cells in the presence or absence of PHIP and showed convincing evidence that in the absence of PHIP there is recruitment of NuRD to promoters and transcriptional repression of genes involved in cell proliferation and cell cycle progression, whereas in the presence of PHIP some NuRD subunits are ubiquitinated and degraded, thus allowing the transcription of genes important for proliferation. The authors also showed that the same mechanism can be observed for SCCOHT and propose that this is a conserved mechanism for cancers with broadly impaired SWI/SNF function. The authors also showed that re-introduction of SMARCB1 in RT cells re-established proper enhancer architecture and caused a redistribution of NuRD complexes to enhancers, whereas in cancers with disrupted SWI/SNF function RT cells depend on gene promoters for transcription and thus recruitment of NuRD to promoters silences these promoters and PHIP is required to ubiquitinate and degrade NuRD and allow transcription of these genes. The authors also confirmed all these results using an in vivo model AT/RT tumors organoids where they conclusively showed that these tumors depended on PHIP, whereas other tumors do not. This manuscript is clearly written, their conclusions are supported by the data and I recommend it for publication without any further changes.

Reviewer #2

(Remarks to the Author)

The manuscript presents a model in which PHIP acts as a promoter-bound scaffold to recruit CRL4, which ubiquitinates the NuRD complex (mainly CHD4), driving NuRD degradation and lowering histone acetylation. This mechanism is relevant in SWI/SNF-deficient tumors—primarily rhabdoid tumors (RTs) and SCCOHT—while intact SWI/SNF complexes in normal cells largely counteract it.

Strengths

The data is of high-quality, with well-executed experiments. Effects are sometimes modest but likely reflect biological subtleties. This is a clear "bottom-up" study design (genes → tumor), typical of CRISPR KO-driven studies. The manuscript is easy to follow despite technical complexity; Figure 7 is particularly effective in summarizing results. Identification of PHIP as a novel vulnerability in SWI/SNF-deficient tumors, with convincing in vivo preclinical data (albeit limited to ATRT). While PHIP has been implicated in other cancers (melanoma, breast, lung, AML), this work provides the first detailed mechanistic

description. The work will be of high interest in both the cancer and epigenetic fields.

Weaknesses (not prohibitive for publication)

- The stepwise introduction of players (PHIP, CRL4, NuRD) improves clarity but obscures why these specific factors were prioritized.
- Claims of universality across all SWI/SNF-deficient cancers are overstated. Strong data come from MRT (G401), partially supported in ATRT and SCCOHT lines. In vivo validation is limited to two ATRT models. Frequent switching between models weakens the generality of conclusions. The SMARCB1-rescue experiment in 293T cells is too artificial to support broad claims.
- The mechanistic chain (PHIP–CRL4–NuRD–acetylation) and its effect on growth/tumorigenesis are shown, but the gene-level link is missing: which genes are directly regulated by this axis in SMARCB1-deficient cells? GO analysis (Ext. Fig. 2I–K) only partially addresses this. Direct identification of key growth-supporting genes across MRT, ATRT, and SCCOHT would strengthen the story and complement recent MYC ATRT (PMID 40121526) and chordoma (PMID 40481032) studies.
- All mechanistic work is cell line–based. ChIP-seq in PDOX tumors or tumoroids would have reinforced conclusions (or could be considered for follow-up work).

Line-by-line / figure-specific comments

- Fig. 1A (line 97): please provide a shortlist of top hits and clarify why PHIP was prioritized.
- Fig. 1B: explain why SCCOHT cells appear more PHIP-sensitive in DepMap. Could SMARCB1 loss (RT) vs. SMARCA4 loss (SCCOHT) account for this?
- Fig. 1C: include SCCOHT BIN16 line.
- Fig. 1D: clarify “normalized time” and show raw data.
- Fig. 2A: PHIP appears at promoters marked by both H3K4me3 and H3K27me3. Are these bivalent? More detail on PHIP binding to K27me3 sites would help.
- Line 123: show ATRT and SCCOHT data if available.
- Fig. 2E: show HA-PHIP ChIP heatmap after siPHIP knockdown to assess residual binding.
- Lines 138–139: clarify whether H3K4me3 status also changes.
- Fig. 3A: IP experiments would be more convincing than chromatin fractionation to prove PHIP–CRL4 interaction.
- Ext. Fig. 3B: why A204 cells instead of G401? Consistency would strengthen conclusions. Show ATRT/SCCOHT data if available.
- Lines 166–167: justify focus on NuRD in IP-MS; why not other enriched proteins?
- Fig. 3D: clarify whether PHIP remains bound to CRL4 during purification, and address specificity of ubiquitination. More negative controls would strengthen claims. Do authors detect ubiquitinated NuRD members by western blot in SWI/SNF-mutant lines?
- Fig. 3E: effect is subtle. Could be strengthened with reverse IPs, other CRL4 subunits, or another RT line.
- Fig. 3F: expand caption—unclear as presented.
- Fig. 4C / Ext. Fig. 4B–C: NuRD recruitment effects are subtle. Additional data in another line or using other histone acetylation marks would help. Enlarging Ext. Fig. 4 panels would improve readability.
- Ext. Fig. 4B–C: authors should clarify how different NuRD subunits behave differently on chromatin. Does the complex assemble in situ?
- Fig. 4F: histone mark ratios are subtle. Orthogonal confirmation (e.g. ChIP-seq) would be stronger.
- Lines 251–253: correlate promoter/enhancer changes upon SMARCB1 rescue with RNA-seq data to link chromatin changes to gene activation. Highlight key genes.
- Line 274: indicate whether ATRT models tested belong to Shh, Tyr, or Myc subtype, as this may affect comparability with G401.
- Lines 326–328: discuss possible toxicities of systemic PHIP inhibition, given its expression in adult tissues.

Typos / minor corrections

- Line 125: should reference Fig. 2B,C (not 2C,D).
- Ext. Fig. 2E: only siRNA B shown; figure title mislabeled. Clarify comparison intended.
- Fig. 6A: “NGS” mouse should read “NSG”.

Reviewer #3

(Remarks to the Author)

PHIP (also known as DCAF14) is a DDB1-CUL4 adaptor reported to recognize acetylated histones and tether CRL4 to chromatin (PMID: 34819353). While PHIP's roles in DNA replication have been described (PMIDs: 27272143, 30018425, 33503431), Malone, Myers and colleagues propose a new function for PHIP in transcriptional activation. They identify PHIP as a selective dependency in cancers with SWI/SNF loss, show that PHIP recruits CRL4 to chromatin, and present evidence that this promotes degradation of the NuRD subunit CHD4, an effect heightened in SMARCB1-deficient cells, thereby protecting genes from silencing. They further test the model in patient-derived orthotopic xenografts and suggest PHIP inhibition as a therapeutic strategy in SWI/SNF-deficient tumors.

The study is potentially interesting, however, the central mechanistic claim that CHD4 is a bona fide substrate of CRL4-PHIP relies heavily on correlative data and lacks critical genetic rescue and biochemical validation.

Major Concerns:

Fig. 1; Please, verify PHIP dependency with at least two independent gRNAs per gene in 3-4 RT cell lines. To assess that the growth phenotype is PHIP dependent, perform rescue by re-expressing PHIP(WT) in PHIP-deficient cells to restore

proliferation. As a specificity control for E3 function, authors can use a DDB1-binding-deficient PHIP mutant (i.e., CRL4 recruitment defective) to test whether CRL4-PHIP ligase activity is required for RT cell proliferation.

Ext. Fig. 2E; RNA-seq was performed after siRNA knockdown of PHIP, which poses a risk for off-target effects. If transient depletion is essential, include siRNA-resistant PHIP cDNA rescue to show that PHIP-dependency of differentially expressed genes.

Fig. 2E; as in the point above, ChIP-seq profiling under siRNA depletion should be accompanied by genetic rescue to show that losses of H4ac (and other marks such as H3K27ac/H3K14ac) at PHIP-regulated loci are PHIP-specific.

Fig. 3A-B; Given that DDB1 and CUL4 are shared across many CRL4 substrate receptors on chromatin, their broad chromatin loss upon siPHIP is surprising and could reflect off-target effects. Please include PHIP re-expression rescue to restore chromatin-bound DDB1/CUL4 and, if possible, acute auxin-degron or dTAG approaches to minimize adaptation.

Fig. 3C; Validate CHD4/MBD3/MTA2/HDAC1/RBBP4 IP-MS interactions by IP-immunoblot. Given the peptide abundance reported, these confirmations should be straightforward.

Fig. 3D; Clarify the assay: it appears PHIP-FLAG immunocomplexes were incubated with E1/E2 and endogenous CHD4/MBD3/MTA2 ubiquitination was assessed. If that is the case, provide evidence that the CHD4 smear represents polyubiquitination. Also, reconstitution of fully defined *in vitro* ubiquitination with recombinant CRL4-PHIP and recombinant CHD4 (and other NuRD subunits as controls) to test direct substrate status is required.

Based on data in Fig. 3D, it is assumed that CHD4 is the direct target of PHIP. Can the authors determine whether CHD4 is the direct PHIP interactor within the NuRD using *in vitro* binding assay?

Fig. 3F; PHIP depletion reduces CHD4/HDAC1/2/RBBP4/MBD3 in the NuRD co-eluting fractions. If PHIP promotes CHD4 degradation, one might expect increased CHD4, not reduced signal in the NuRD fractions.

Fig. 4A, 4C; PHIP overexpression causes only modest decreases in chromatin-bound CHD4/NuRD, and the reported log₂ fold-changes on H4ac sites upon siPHIP are only ~0.2-0.3; small effects for a primary degradation mechanism. Can the authors test pathway dependence with proteasome inhibition (e.g., MG132/bortezomib) and neddylation inhibition (MLN4924/pevonedistat) to determine whether the observed changes are proteasome- and cullin-dependent.

Fig. 6B, 6D; Are the phenotypes caused by PHIP loss rescued by co-depletion of CHD4 (or stabilized CHD4)? Epistasis would strongly support CHD4 as the relevant substrate/effector in this context.

Minor Concerns:

Lines 56-58: "SMARCB1" should be SMARCB1. Also, the blanket statement that SMARCB1 and SMARCA4 are "bona fide tumor suppressors" is imprecise; SMARCA4 can act as an oncogene in certain contexts (e.g., breast, prostate, colorectal; PMID: 36769189).

Many main-figure analyses use HA-PHIP ChIP-seq from G401 cells overexpressing HA-PHIP. This should be explicitly noted in the main text (not only legends), with rationale for choosing HA-PHIP over endogenous profiling and a discussion of potential overexpression confounders.

Version 1:

Reviewer comments:

Reviewer #1

(Remarks to the Author)

The authors have done extensive studies to address the concerns of the other reviewers and the manuscript in its present form is stronger and a more rigorous study.

Reviewer #2

(Remarks to the Author)

The authors adequately addressed all my comments. This already strong manuscript is further strengthened by the data and clarifications included. I unreservedly recommend publication.

Reviewer #3

(Remarks to the Author)

The authors have been exceptionally thorough in addressing my previous concerns. The revisions are comprehensive and significantly enhance the quality of the manuscript. I have no further questions or comments.

We sincerely thank the editors and reviewers for the time, care, and expertise they devoted to evaluating our manuscript. We were pleased to hear the reviewers' assessment that our data "is of high-quality, with well-executed experiments" and that the study "will be of high interest in both the cancer and epigenetic fields" and is "well written with conclusions supported by the data."

We greatly appreciate the insightful comments and constructive suggestions from the reviewers, who raised important matters that have enabled us to substantially improve the clarity, rigor, and impact of the manuscript. In response to this feedback, we have performed extensive new functional and mechanistic experiments, the results of which further substantiate our central conclusion that PHIP functions as an essential suppressor of NuRD complexes in the context of SWI/SNF deficiency. These additions include critical rescue experiments demonstrating the on-target function and dependency of PHIP, as well as expanded biochemical analyses validating NuRD as a direct target of PHIP. We have revised the manuscript to include these new data and address the review comments.

Specifically, to address the reviewers' concerns and strengthen our proposed model, we have

- **Mitigated concerns regarding off-target dependency** by repeating growth assays with additional independent sgRNAs and by analyzing raw DepMap data, demonstrating that all four PHIP-targeting guides score as a selective dependency.
- **Established on-target PHIP activity** in regulating transcription, histone acetylation, cellular proliferation, and CRL4 recruitment across gain-of-function, loss-of-function, and rescue conditions.
- **Demonstrated that CRL4–PHIP ligase activity is required for rhabdoid tumor cell proliferation** by using a PHIP mutant defective in CRL4 recruitment.
- **Identified CHD4 as a direct substrate of PHIP-mediated ubiquitination** through additional biochemical assays.
- **Shown that PHIP-mediated antagonism of NuRD is cullin-dependent**, using complementary genetic and pharmacological approaches.

Together, these new data reinforce our original findings that establish PHIP as a specific vulnerability in subsets of SWI/SNF-mutant cancers, as well as a critical regulator of NuRD–SWI/SNF antagonism through a previously unappreciated mechanism of ubiquitin-mediated suppression of NuRD function. We are grateful to the reviewers for their thoughtful critiques, which prompted experiments that significantly strengthened both the mechanistic depth and the overall impact of the study.

Below, we provide point-by-point responses to each of the reviewers' comments.

REVIEWER COMMENTS

Reviewer #1 (Remarks to the Author):

The manuscript by Malone et al. entitled "PHIP suppresses NuRD to enable the growth of SWI/SNF-mutant cancers" the authors identified PHIP as a dependency for cancers with disrupted SWI/SNF function. SWI/SNF chromatin remodelers are large multi-subunit complexes mutated in up to 20% of cancers and thus the identification of a vulnerability in these SWI/SNF-mutated cancers is an important advance in the field and has therapeutic implications for the treatment of SWI/SNF-mutated cancers. The

authors identified PHIP through a CRISPR/Cas9 screen that compared dependencies between cancer cells, Rhabdoid tumors (RT), and small cell carcinomas of the ovary, hypercalcemic type (SCCOHT). PHIP showed as a dependency on both RTs and SCCOHT, but not in all other cancer types. The authors validated this finding in 293T cells showing that inactivation of SMARCB1 and PHIP resulted in reduced growth, but not by inactivating these genes independently. Using RT cell lines, the authors downregulated PHIP and showed that the inactivation of this factor resulted in downregulation of gene expression of various pathways associated with cell proliferation, thus explaining why RT cells would require this factor for growth. They also showed that PHIP can be found on the promoters of the genes that are downregulated upon PHIP inactivation. The authors found PHIP in a complex with ubiquitin ligases and did IP-MS to identify substrates, which resulted in the identification of the NuRD complex as a target. The authors did a series of experiments using ChIP-Seq in RT cells in the presence or absence of PHIP and showed convincing evidence that in the absence of PHIP there is recruitment of NuRD to promoters and transcriptional repression of genes involved in cell proliferation and cell cycle progression, whereas in the presence of PHIP some NuRD subunits are ubiquitinated and degraded, thus allowing the transcription of genes important for proliferation. The authors also showed that the same mechanism can be observed for SCCOHT and propose that this is a conserved mechanism for cancers with broadly impaired SWI/SNF function. The authors also showed that re-introduction of SMARCB1 in RT cells re-established proper enhancer architecture and caused a redistribution of NuRD complexes to enhancers, whereas in cancers with disrupted SWI/SNF function RT cells depend on gene promoters for transcription and thus recruitment of NuRD to promoters silences these promoters and PHIP is required to ubiquitinate and degrade NuRD and allow transcription of these genes. The authors also confirmed all these results using an in vivo model AT/RT tumors organoids where they conclusively showed that these tumors depended on PHIP, whereas other tumors do not. This manuscript is clearly written, their conclusions are supported by the data and I recommend it for publication without any further changes.

We thank the reviewer for their detailed assessment of our study and for their positive feedback regarding the novelty and quality of this manuscript.

Reviewer #2 (Remarks to the Author):

The manuscript presents a model in which PHIP acts as a promoter-bound scaffold to recruit CRL4, which ubiquitinates the NuRD complex (mainly CHD4), driving NuRD degradation and lowering histone acetylation. This mechanism is relevant in SWI/SNF-deficient tumors—primarily rhabdoid tumors (RTs) and SCCOHT—while intact SWI/SNF complexes in normal cells largely counteract it.

Strengths

The data is of high-quality, with well-executed experiments. Effects are sometimes modest but likely reflect biological subtleties. This is a clear “bottom-up” study design (genes → tumor), typical of CRISPR KO-driven studies. The manuscript is easy to follow despite technical complexity; Figure 7 is particularly effective in summarizing results. Identification of PHIP as a novel vulnerability in SWI/SNF-deficient tumors, with convincing in vivo preclinical data (albeit limited to ATRT). While PHIP has been implicated in other cancers (melanoma, breast, lung, AML), this work provides the first detailed mechanistic description. The work will be of high interest in both the cancer and epigenetic fields.

Weaknesses (not prohibitive for publication)

- The stepwise introduction of players (PHIP, CRL4, NuRD) improves clarity but obscures why these specific factors were prioritized.

1. We thank the reviewer for this important point, and in hindsight we realize that we insufficiently described why these factors were the key focus. We have now edited the manuscript to address this. As background, when we performed experiments to identify genes that are specifically essential in rhabdoid tumors (RTs and AT/RTs), as compared to all other cell lines, we prioritized hits that were 1) highly statistically significant; 2) validated as a dependency that was specifically caused by the absence of SMARCB1 rather than reflecting a dependency of the lineage of origin; 3) had the potential to provide mechanistic insight; and 4) had the potential to be therapeutically actionable. PHIP met all of these criteria. Notably, the statistical significance of dependency upon PHIP exceeds that of EZH2, which we had previously identified as a dependency in RTs, and which culminated in FDA approval for the use of an EZH2 inhibitor to treat SMARCB1-mutant cancers. Mechanistically, PHIP had been linked to chromatin regulation, but the mechanism remained unclear (PMID 29089422, 34819353), as did its relationship to SWI/SNF. With respect to targetability, PHIP contains a unique bromodomain that has been shown to be chemically tractable. Lastly, we established that dependency upon PHIP was caused by the absence of SMARCB1.
2. Our focus on CRL4 was based on previous reports that PHIP was a DCAF protein that coordinated with this E3 ligase, and this was validated in our experiments.
3. When examining targets that were degraded by PHIP, we focused on NuRD because 1) numerous NuRD subunits were detected as interactors of PHIP, and NuRD had been shown to oppose SWI/SNF activity; 2) PHIP knockdown led to de-acetylation of histones, a known activity of NuRD; and 3) as PHIP, part of the ubiquitin ligase pathway, facilitates transcriptional activation, we reasoned that it may be degrading a transcriptional repressor.

We have now clarified each of these aspects in the manuscript.

- Claims of universality across all SWI/SNF-deficient cancers are overstated. Strong data come from MRT (G401), partially supported in ATRT and SCCOHT lines. In vivo validation is limited to two ATRT models. Frequent switching between models weakens the generality of conclusions. The SMARCB1-rescue experiment in 293T cells is too artificial to support broad claims.

We thank the reviewer for bringing this important point to our attention. Vulnerabilities identified in DepMap can sometimes reflect a dependency related to the cell of origin (e.g., a lineage-specific transcription factor) rather than a synthetic lethality conferred by the mechanism underlying oncogenic transformation. To rigorously evaluate whether dependency upon PHIP was a direct result of SWI/SNF perturbation, we used cell lines from RTs of multiple tissues (e.g., MRT of kidney, MRT of liver, and AT/RT) and from SCCOHT. The latter is a cancer driven by perturbation of another SWI/SNF subunit (SMARCA4 rather than SMARCB1); that arises from a different tissue (the ovary vs. the kidney, liver, and brain in RTs) and in patients of a different age (young adults vs. infants) (see Fig. 1D and Extended Data Fig. 1D–G). To further build rigor in this evaluation, we tested whether the dependency upon PHIP was a direct result of SMARCB1 loss by using an unrelated cancer cell line (293T cells) in which we genetically engineered the absence of SMARCB1 (see Fig. 1E). PHIP was validated in all of these models, thus demonstrating that dependence upon PHIP was indeed a direct result of SWI/SNF mutation.

As is the case for dependency upon EZH2 (which ultimately achieved FDA approval as a therapeutic target for SMARCB1-mutant cancers), PHIP does not score as a significant dependency in all SWI/SNF-mutant cancers. In the case of PHIP, dependence is conferred by mutations that strongly and broadly impair the function of multiple SWI/SNF subfamilies. It is for this reason that we had stated in the abstract that PHIP is a dependency “in certain families of SWI/SNF-mutant cancers” and in the text “more limited impairment of SWI/SNF function, such as in cancers with mutations that affect only one member of a paralog pair (e.g., in *ARID1A*) or only one SWI/SNF subfamily (e.g., in *PBRM1*), do not result in enhanced sensitivity to PHIP loss (Extended Data Fig. 1C).” We have revised the text to further amplify this point. Lastly, as *in vivo* experiments are time-consuming and expensive, we felt that expanding the study to include patient-derived tumor organoid models, which are actively used by our collaborators in Dr. Roussel’s laboratory, would be sufficient to confirm that the dependency extends from our cell lines to *in vivo* conditions.

- The mechanistic chain (PHIP–CRL4–NuRD–acetylation) and its effect on growth/tumorigenesis are shown, but the gene-level link is missing: which genes are directly regulated by this axis in SMARCB1-deficient cells? GO analysis (Ext. Fig. 2I–K) only partially addresses this. Direct identification of key growth-supporting genes across MRT, ATRT, and SCCOHT would strengthen the story and complement recent MYC ATRT (PMID 40121526) and chordoma (PMID 40481032) studies.

We thank the reviewer for raising this important point regarding the conserved effects of SMARCB1 loss across the cancer models.

We consider it important to note that SWI/SNF is bound at most active enhancers and is part of the core machinery that facilitates transcriptional activation. Mechanistically, we have previously demonstrated that the loss of SMARCB1 results in broad impairment of enhancer function. In most cell types, loss of SMARCB1 results in either cell death or stasis (due to failed transcriptional programs). However, in susceptible highly proliferative progenitor cells, the impairment of enhancer function results in the inability to activate genes required for differentiation. What is shared across cancers driven by SMARCB1 loss is not perturbed expression of a universally shared set of downstream target genes but rather the failure to activate enhancers that are required for differentiation. These targets are lineage-specific. For example, when we previously compared the genes activated by re-expression of SMARCB1 in kidney RT cell lines vs. brain RT (AT/RT) cell lines vs. liver RT cell lines, we found very little overlap¹. Kidney pathways were activated in the kidney models, neural/brain pathways were activated in the brain models, etc. Although the individual direct targets vary in the different lineages, we do find that genes associated with the cell cycle are secondarily affected, as the re-expression of SMARCB1 in SMARCB1-deficient cells causes the cells to lose proliferative capacity². These findings are consistent with those in the paper by Nesvick et al. (PMID 40121526), which, in its title, states that SWI/SNF is required for “Ontology-specific transcription factor function.”

In the current manuscript, we have identified PHIP as a factor that cooperates with SWI/SNF to facilitate gene expression and that becomes essential for transcription when SWI/SNF function is broadly inactivated. Consistent with the findings of Nesvick et al. (PMID 40121526) and Liu et al. (PMID 40481032), we observed that the loss of PHIP led to a reduction in the expression of cell cycle genes essential for viability. We have sought to highlight this in Extended Data Figure panels 2J and K.

Top 250 downregulated by PHIP KD (G401)

Propidium iodide (PI) cell cycle analysis

To address the reviewer's query further, we have now extended our gene expression analysis to two additional cell lines, the AT/RT cell line CHLA266 and the SCCOHT cell line BIN67. Performing GO analysis in the collective models again highlighted changes in the cell cycle (as in PMID 40121526 and PMID 40481032), as would be expected from the loss of core transcription function. These results are now included in Extended Data Figure 2L and M.

A204 (RNA-seq)

BIN67 (RNA-seq)

CHLA266 (RNA-seq)

Top 250 downregulated by PHIP KD (A204)

Top 250 downregulated by PHIP KD (BIN67)

Top 250 downregulated by PHIP KD (CHLA266)

- All mechanistic work is cell line-based. ChIP-seq in PDOX tumors or tumoroids would have reinforced conclusions (or could be considered for follow-up work).

Thank you for this recommendation. Although we include ChIP-seq data from just one RT cell line, we show that PHIP also recruits CRL4, suppresses NuRD, and preserves histone acetylation by subcellular fractionation in both RT and SCCOHT cell lines. We agree that extending this investigation to PDOX tumors or tumoroids could be considered for follow-up work.

Line-by-line / figure-specific comments

- Fig. 1A (line 97): please provide a shortlist of top hits and clarify why PHIP was prioritized.

Thank you for raising this important point. Based on your recommendation, we now include a short list of top hits as Extended Data Figure 1B. We have added a description of our rationale for prioritizing PHIP above, and we have added additional text to the manuscript to clarify this. We now specify that PHIP, as an RT-specific vulnerability, scores as more significant than several targets that are being pursued in

clinical trials (MDM4³, BRD9^{4,5}) and one target (EZH2⁶) that has received FDA approval for SMARCB1-mutant cancers.

Gene	Effect	-log10p
NABP2	-0.509	20.3
DCAF5	-0.261	9.64
PHIP	-0.337	8.94
EZH2	-0.406	6.59
MDM4	-0.433	5.00
BRD9	-0.319	4.85

- Fig. 1B: explain why SCCOHT cells appear more PHIP-sensitive in DepMap. Could SMARCB1 loss (RT) vs. SMARCA4 loss (SCCOHT) account for this?

This is an interesting observation about which we have also been curious. SMARCB1 is a member of the two most dominant/active SWI/SNF subfamilies (CBAF and PBAF) but not of the third (ncBAF), which contributes less strongly to transcriptional regulation. We have previously demonstrated that ncBAF is required for cell survival when SMARCB1 is absent, indicating that it has some activity in this setting. SCCOHT cells have inactivating mutations in SMARCA4 and do not express SMARCA2, meaning that the activity of all three SWI/SNF subfamilies is abolished. As there is no residual SWI/SNF function in these cells, we speculate that this may explain why PHIP dependency is even greater in SCCOHT cells.

- Fig. 1C: include SCCOHT BIN16 line.

We agree that adding the SCCOHT cell line will increase the robustness and breadth of the analyses, and we thank the reviewer for the suggestion. We have now included results from a CelFi assay in BIN67 cells in Figure 1D. Similar to the RT cell lines included in the original analysis, BIN67 cells select against damaging out-of-frame PHIP mutations.

- Fig. 1D: clarify “normalized time” and show raw data.

In the original figure, we used “Normalized time” (elapsed time/total time recorded) to compare the effect of PHIP knockdown on cell lines that have differing proliferation rates. In response to the reviewer’s request, we have updated Figure 1E to include a panel for each cell line. This change does not alter the interpretation of the results.

- Fig. 2A: PHIP appears at promoters marked by both H3K4me3 and H3K27me3. Are these bivalent? More detail on PHIP binding to K27me3 sites would help.

The reviewer raises an important point regarding bivalency that we had not previously evaluated. Because a subset of PHIP peaks do appear to co-localize with H3K27me3 in Figure 2A, we asked whether PHIP binding was frequently detected at H3K27me3-marked sites in RT cells. We generated a heatmap of H3K27me3 peaks and plotted the ChIP-seq signal for PHIP, H3K27ac, H3K14ac, and H3K4me3 at these sites (see Extended Data Figure 2C). As H3K27me3 peaks are broad, we scaled the peaks to a uniform width by using the scale-regions function in deepTools. When the heatmap is sorted by PHIP signal strength, the data show that PHIP is bound at only a small subset of H3K27me3 peaks, and these sites are indeed the fraction that also have H3K4me3. Ultimately, PHIP is bound at both active promoters (positive for both H3K4me3 and H3K27ac but negative for H3K27me3) and bivalent promoters (those also positive for H3K27me3).

- Line 123: show ATRT and SCCOHT data if available.

Thank you for this suggestion. We have completed RNA-seq after PHIP knockdown in a SCCOHT cell line (BIN67) and an AT/RT cell line (CHLA266). The results from this are shown in Extended Data Figure 2L and M. Similar to the RT cell lines included in the initial submission (G401 and A204), we observed an enrichment of downregulated genes after PHIP knockdown in SCCOHT and AT/RT cell lines, and silenced genes were enriched for pathways controlling replication and the cell cycle.

- Fig. 2E: show HA-PHIP ChIP heatmap after siPHIP knockdown to assess residual binding.

We agree that it is important to consider the effect of residual PHIP function after knockdown. In Extended Data Figure 2A, we show controls for both the antibody to HA and the antibody to endogenous PHIP. For the antibody to HA, we used a condition in which the cells had been transduced with HA-GFP. For the antibody to endogenous PHIP, as requested, we show the effect of PHIP knockdown on peak detection. In both cases, the signal is virtually ablated in the control condition.

- Lines 138–139: clarify whether H3K4me3 status also changes.

Thank you for this suggestion. We have now included the results of ChIP-seq for H3K4me3 after PHIP knockdown at the same loci originally included in Figure 2F. Unlike histone acetylation, we observed no reductions in H3K4me3. The lack of change is consistent with NuRD complexes having deacetylase activity but not demethylase activity.

- Fig. 3A: IP experiments would be more convincing than chromatin fractionation to prove PHIP–CRL4 interaction.

In response to the reviewer's concern, we have performed IP-MS and Co-IP experiments, and we now include results that confirm the PHIP–CRL4 interaction. These results are shown in Extended Data Figures 3B–D.

- Ext. Fig. 3B: why A204 cells instead of G401? Consistency would strengthen conclusions. Show ATRT/SCCOHT data if available.

We apologize for any confusion that may have arisen from the data for G401 and A204 cells being presented in different figures. To maintain the main figures at an acceptable size for publication, we included the G401 data in Main Figure 3A and the A204 data in the supplemental data (Extended Data Figure 3A). We have now improved the citation of these panels to make this clearer.

To further address the reviewer's request for consistency to strengthen our conclusions, we have now added the CRL4 subunit DDB1 to Figure 5E, which demonstrates that PHIP also recruits CRL4 to chromatin in non-RT cells. In Figure 4H, we show that this role is also conserved in SCCOHT cell lines.

- Lines 166–167: justify focus on NuRD in IP-MS; why not other enriched proteins?

We thank the reviewer for raising this important point. In hindsight, we realize that we failed to communicate fully the rationale for the focus on NuRD. We focused on the NuRD complex for several reasons. First, 10 different components of NuRD were enriched in the IP-MS experiment. In the raw data included in the original submission, many of the proteins represented common contaminants in IP-MS experiments. Because PHIP localizes exclusively to chromatin (based on the subcellular fractionation experiments in Figure 3A), we have now filtered the list of IP-MS hits to include only chromatin-associated proteins (using the Chromatin GO:0000785 gene list). We have updated Figure 3F to display this filtered plot, which shows that 10 out of 75 interactors with PHIP are subunits of the NuRD complex.

Second, our early experiments had revealed a reduction in histone acetylation after PHIP inactivation, although the mechanism was unknown. Because PHIP cooperates with the E3 ubiquitin ligase machinery to ubiquitinate and potentially degrade proteins, identifying the histone deacetylase known to erase these marks (NuRD) provided a compelling mechanistic link. Third, antagonism between SWI/SNF and NuRD complexes has been reported in development and in disease, so it was plausible that a negative regulator of the NuRD complex might become critical when SWI/SNF was impaired. In response to the reviewer's feedback, we have now expanded upon the logic in the first and fourth paragraphs of the section titled "PHIP recruits E3 ligases to chromatin and ubiquitinates NuRD."

- Fig. 3D: clarify whether PHIP remains bound to CRL4 during purification, and address specificity of ubiquitination. More negative controls would strengthen claims. Do authors detect ubiquitinated NuRD members by western blot in SWI/SNF-mutant lines?

Thank you for the opportunity to clarify this assay and result. CRL4 remains bound to PHIP during purification, as evidenced by the detection of the CRL4 subunit DDB1 by Western blot in samples that include FLAG-PHIP eluate. The lack of DDB1 ubiquitination demonstrates that PHIP is not promiscuously ubiquitinating every protein in the assay. To demonstrate that the size shift in the CHD4 blot accurately represents ubiquitination, we have repeated this experiment with an additional negative control, and we now show that the size shift does not occur when ubiquitin is omitted from the reaction (see Figure 3G). Additionally, our results demonstrate that PHIP exhibits selectivity within NuRD complexes, as evidenced by the minimal ubiquitination of RBBP4.

To understand how PHIP targets NuRD complexes, we have now investigated whether PHIP can recognize and ubiquitinate recombinant subunits of the NuRD complex (CHD4 and 6X-His-HDAC1) in *in vitro* assays. We found that PHIP can directly recognize and ubiquitinate solo CHD4 but not HDAC1, as is now shown in Extended Data Figure 3I. These results suggest that CHD4 is sufficient for recognition and ubiquitination by PHIP.

We detected no ubiquitinated NuRD subunits in RT cells by Western blot. Because PHIP co-localizes with NuRD at only a minority of NuRD sites (see Fig. 4C), we were not surprised that it was challenging to detect global changes in ubiquitination after PHIP loss.

- Fig. 3E: effect is subtle. Could be strengthened with reverse IPs, other CRL4 subunits, or another RT line.

Thank you for this recommendation. To the original figure, which showed only the results for CRL4 subunit DDB1, we have now added the results for CUL4A. This revised figure is now included as Figure 3J and is reproduced below for ease of review. The changes in CUL4A parallel those seen in DDB1, thus reinforcing the original conclusions.

- Fig. 3F: expand caption—unclear as presented.

We have now expanded the caption to clarify the content.

- Fig. 4C / Ext. Fig. 4B–C: NuRD recruitment effects are subtle. Additional data in another line or using other histone acetylation marks would help.

As PHIP only co-localizes with NuRD at a minority of NuRD sites (shown in Figure 4C), we expect PHIP to influence NuRD turnover only at this subset. For this reason, it is not surprising that Extended Data Figure 4D shows that only a fraction of CHD4 peaks gain binding when PHIP is inactivated.

Nevertheless, orthogonal approaches show clear losses of histone acetylation after PHIP inactivation (by Western blot in three RT cell lines in Figure 2E, by unbiased histone-MS in Extended Data Figure 2H and I). In response to the Reviewer’s critique, to evaluate the on-target activity of PHIP in controlling histone acetylation, we have now performed H3K27ac ChIP-seq after overexpression, knockdown, or genetic rescue of PHIP, and we present these results in Figure 2G, reproduced below for convenience. These findings demonstrate that the control of histone acetylation by PHIP is indeed on-target.

Enlarging Ext. Fig. 4 panels would improve readability.

Thank you for bringing this to our attention. We have now enlarged the panels.

- Ext. Fig. 4B–C: authors should clarify how different NuRD subunits behave differently on chromatin. Does the complex assemble *in situ*?

Thank you for the opportunity to clarify this. Nucleosome remodeling and deacetylase (NuRD) complexes are multi-subunit chromatin-modifying complexes that silence transcription by combining two enzymatic activities: nucleosome remodeling through CHD3, CHD4, or CHD5 and histone deacetylation through HDAC1 or HDAC2⁷⁻⁹. Structurally, NuRD complexes are assembled onto chromatin *in situ* as two distinct modules responsible for each enzymatic function¹⁰. The core deacetylase module contains HDAC1/2, histone chaperones RBBP4/7, and chromatin-binding and scaffolding proteins MTA1/2/3, and it can independently bind to chromatin as a stable subcomplex with detectable enzymatic activity¹¹. CHD4 binds to chromatin and forms a second subcomplex with GATAD2A/B and CDK2AP1. MBD2/3 directly interacts with the remodeling (CHD4) and deacetylase (HDAC) subcomplexes and bridges their function to constitute full NuRD complexes¹². These two distinct modules are functionally linked, as remodeling by CHD4 enhances deacetylation of nucleosomes by HDAC1/2¹³⁻¹⁵. We have now included this description in the text.

- Fig. 4F: histone mark ratios are subtle. Orthogonal confirmation (e.g. ChIP-seq) would be stronger.

We thank the reviewer for this recommendation. We agree that the reductions in histone acetylation after PHIP knockdown are subtle in the referenced figure. During the revision process for this manuscript, we have devoted significant effort to determining whether changes in histone acetylation and transcription are truly PHIP-dependent. As discussed above, we have now performed ChIP-seq for H3K27ac upon PHIP overexpression, PHIP knockdown, and PHIP rescue conditions. The results, now included in Figure 2G, demonstrate the on-target activity of PHIP in controlling histone acetylation.

Based on your recommendation, we additionally performed ChIP-seq for H4ac in control, PHIP KD, CHD4 KD, and double-knockout conditions. Unfortunately, the data from this experiment were of marginal quality. Outliers in the data prevented robust analysis, but some trends were still clear. We saw an enrichment of sites where acetylation is lost upon PHIP KD and gained upon CHD4 KD, consistent with the observations in the figure you referenced (now Figure 4G). However, we are hesitant to include these data or to make any conclusions about the double-knockout condition, given the limitations of this marginal-quality dataset. As your feedback indicated that this weakness is not prohibitive to publication, and because our new H3K27ac results presented in Figure 2G reinforce our conclusions, we decided to submit the revised manuscript without this ChIP-seq data within the 3-month time frame requested by the editor, rather than requesting an extension of the deadline.

- Lines 251–253: correlate promoter/enhancer changes upon SMARCB1 rescue with RNA-seq data to link chromatin changes to gene activation Highlight key genes.

Thank you for bringing this important point to our attention. Previously, our lab and others have demonstrated that the rescue of SMARCB1 in RT cells restores enhancer function and activates gene expression. Specifically, it rescues lineage-specific enhancers that control the expression of genes controlling differentiation^{1,16}. SMARCB1 rescue does not activate a shared set of genes between RT cell lines derived from distinct lineages; instead, it activates differentiation pathways relevant to each lineage. Consequently, the data suggest that the tumor suppressor activity of SMARCB1 is not derived from universal control of a set of “growth/anti-growth” genes but rather from facilitating activation of lineage-specific gene expression pathways. In select highly proliferative progenitor cells, loss of the ability to activate enhancers that facilitate differentiation appears to be what drives uncontrolled proliferation, and also explains why loss of SWI/SNF genes causes cancers in a highly context-specific manner. We have now added additional citations of these studies in our text, and we thank the reviewer again for bringing this to our attention.

- Line 274: indicate whether ATRT models tested belong to Shh, Tyr, or Myc subtype, as this may affect comparability with G401.

Thank you for the opportunity to clarify this. We have included tumor organoids from distinct AT/RT subtypes: SHH (SJATRT041800) and MYC (SJATRT059003), which are now defined in the text.

- Lines 326–328: discuss possible toxicities of systemic PHIP inhibition, given its expression in adult tissues.

Most chemotherapies used in the clinic are highly toxic across cell types, whereas the consequences of PHIP inactivation appear to be more targeted. We demonstrate that PHIP loss has the strongest effects on cells with mutations that have a severe impact on SWI/SNF complexes and that this specific vulnerability does not extend even to cancers with less serious mutations in SWI/SNF complexes that still allow remodeling (see Extended Data Figure 1C). In Figure 5E, we demonstrate that the ability of PHIP to evict NuRD complexes from chromatin is enhanced when SWI/SNF remodeling is eliminated. We speculate that mutation of SMARCB1 or SMARCA4 results in enhanced dependence upon PHIP as a consequence of the impaired function of the cooperating SWI/SNF complex. Although it is conceivable that systemic PHIP inhibition would have some effect on other cells, our finding of enhanced dependence upon PHIP in cell lines with SMARCB1 or SMARCA4 mutations, as compared to more than 1000 other cell lines, suggests a strong therapeutic window. We have now highlighted this topic in our Discussion, and we thank the Reviewer for the opportunity to expand upon this.

Typos / minor corrections

- Line 125: should reference Fig. 2B,C (not 2C,D).

We have corrected the citation of the referenced figures.

- Ext. Fig. 2E: only siRNA B shown; figure title mislabeled. Clarify comparison intended.

We have clarified the comparison and figure labeling in the referenced figure (now Extended Data Figure 2F). Briefly, we performed GSEA using a ranked list of differentially expressed genes after PHIP knockdown with one siRNA (designated “siPHIP D”), compared to lists of upregulated (left) or downregulated (right) genes after PHIP knockdown with a different siRNA (designated “siPHIP B”). We have updated the labeling on the figure to clarify this, and we thank the reviewer for bringing this to our attention.

- Fig. 6A: “NGS” mouse should read “NSG”.

We thank the reviewer for catching this mistake; it has now been corrected.

Reviewer #3 (Remarks to the Author):

PHIP (also known as DCAF14) is a DDB1-CUL4 adaptor reported to recognize acetylated histones and tether CRL4 to chromatin (PMID: 34819353). While PHIP's roles in DNA replication have been described (PMIDs: 27272143, 30018425, 33503431), Malone, Myers and colleagues propose a new function for PHIP in transcriptional activation. They identify PHIP as a selective dependency in cancers with SWI/SNF loss, show that PHIP recruits CRL4 to chromatin, and present evidence that this promotes degradation of the NuRD subunit CHD4, an effect heightened in SMARCB1-deficient cells, thereby protecting genes from silencing. They further test the model in patient-derived orthotopic xenografts and suggest PHIP inhibition as a therapeutic strategy in SWI/SNF-deficient tumors.

The study is potentially interesting, however, the central mechanistic claim that CHD4 is a bona fide substrate of CRL4-PHIP relies heavily on correlative data and lacks critical genetic rescue and biochemical validation.

Major Concerns:

Fig. 1; Please, verify PHIP dependency with at least two independent gRNAs per gene in 3-4 RT cell lines.

Thank you for this recommendation. We agree that it is essential to minimize the impact of off-target effects when determining whether PHIP is a specific dependency in the cancers we describe. We have addressed your request in the following ways:

1. Our original observation that PHIP is a specific vulnerability in RT and SCCOHT cell lines came from DepMap screens performed using multiple gRNAs from the Avana library. The CHRONOS score, which quantifies the sensitivity of a cell line's sensitivity to gene inactivation, is calculated by integrating scores across individual gRNAs targeting each gene. We have now mined the raw DepMap data to evaluate the scores of individual PHIP gRNAs. There are four guides targeting PHIP, and these target three non-overlapping loci. Our new analysis revealed that all four of the four gRNAs targeting PHIP are selectively disadvantageous in RT/SCCOHT cells when compared to all other cells. These data are now included in Extended Data Figure 1A.

2. Additionally, we performed an orthogonal dependency assay (CelFi¹⁷) using two additional gRNAs targeting PHIP at loci distinct from those targeted by gRNAs in the Avana library. This assay was performed in an MRT cell line (G401) an ATRT cell line (CHLA266), and a now a SCCOHT cell line (BIN67) with a neuroblastoma cell line (Kelly) as a control. Similarly, we observed a selection

against damaging out-of-frame mutations in RT cell lines but not in control lines with intact SWI/SNF, as shown in Figure 1D.

D

- As recommended, we have now repeated proliferation assays using two independent gRNAs in RT cell lines from distinct lineages: two kidney RT cell lines (G401, A204), a liver RT cell line (TTC549), and an AT/RT cell line (CHLA266). In all cases, both guides led to reduced cell proliferation when compared to non-targeting gRNAs. These results are now included in Extended Data Figure 1D and E.

E

To assess that the growth phenotype is PHIP dependent, perform rescue by re-expressing PHIP(WT) in PHIP-deficient cells to restore proliferation. As a specificity control for E3 function, authors can use a DDB1-binding-deficient PHIP mutant (i.e., CRL4 recruitment defective) to test whether CRL4-PHIP ligase activity is required for RT cell proliferation.

Ext. Fig. 2E; RNA-seq was performed after siRNA knockdown of PHIP, which poses a risk for off-target effects. If transient depletion is essential, include siRNA-resistant PHIP cDNA rescue to show that PHIP-dependency of differentially expressed genes.

Fig. 2E; as in the point above, ChIP-seq profiling under siRNA depletion should be accompanied by genetic rescue to show that losses of H4ac (and other marks such as H3K27ac/H3K14ac) at PHIP-regulated loci are PHIP-specific.

As noted by the reviewer, these three points are related in seeking validation that the changes we have identified are indeed consequences of PHIP loss and not off-target effects. We have performed new experiments to address each of the three points, and the results are grouped below. In each case, rescue with PHIP reversed the changes caused by PHIP knockdown, demonstrating that the reported effects were direct consequences of PHIP loss.

With respect to the question regarding the rescue of proliferation, the reviewer has raised two important questions: 1) whether the proliferation phenotype is PHIP-dependent (i.e., an on-target effect that can be rescued by re-expressing PHIP) and 2) whether interaction with CRL4 is essential for this activity. To address this, we developed cells as suggested with DOX-inducible expression of either full-length PHIP or a mutant version of PHIP that cannot interact with CRL4 (Δ H-PHIP¹⁸). The IP-MS results confirmed that

Δ H-PHIP cannot interact with CRL4, and chromatin blots showed that Δ H-PHIP overexpression does not enhance CRL4 recruitment to chromatin. These data are now included in Figure 3B–D.

We next addressed the Reviewer’s question of whether rescue of PHIP expression could rescue the proliferation effects caused by PHIP knockdown and whether this activity was dependent upon CRL4 interaction. We have now shown that the anti-proliferation effects of gRNAs targeting PHIP can be rescued by over-expressing an HA-PHIP construct that is not recognized by these gRNAs (Extended Data Fig. 1H; below, left). Additionally, we found that expression of the HA- Δ H-PHIP mutant was incapable of rescuing the proliferation of PHIP-knockout cells. These findings are now included in Extended Data Figure 3E (below, right).

Collectively, these results establish both that the impaired proliferation of RT cells upon PHIP knockout is an on-target effect and that interaction with CRL4 is essential for this activity.

We next addressed the reviewer’s question about the specificity of transcriptional changes caused by transient PHIP depletion. Based on the Reviewer’s recommendation, we performed rescue RNA-seq experiments using siRNA-resistant PHIP cDNA. We found that, similar to the effects on proliferation, overexpression of PHIP can rescue the transcriptional consequences of siRNA-induced PHIP knockdown. The effects of PHIP upon transcription were consistent and were reflected in a robust group of 2077 genes that were upregulated when PHIP was overexpressed, downregulated when PHIP was knocked out, and rescued in the rescue conditions. These data are now included in Figure 2D.

RNA-seq at PHIP target genes n=2077

These results demonstrate that transcriptional changes are a direct consequence of PHIP loss, rather than an artifact or off-target effect of siRNA treatment. Also supporting this, we have included confirmation (in Extended Data Figure 2E and F) that using independent siRNAs to target PHIP causes similar transcriptional changes.

Finally, to address the reviewer's question regarding the specificity of PHIP as a regulator of histone acetylation, we have now performed ChIP-seq for H3K27ac in the same rescue conditions as requested. We observed that PHIP overexpression led to increases in H3K27ac, PHIP knockdown reduced H3K27ac, and genetic rescue of PHIP restored H3K27ac to near wild-type levels. These results support an on-target function of PHIP in controlling histone acetylation and are now included in Figure 2G.

H3K27ac at lost peaks after PHIP KD LFC<-1, n=22193

Together, our new results confirm that PHIP's role in preserving histone acetylation and activating gene expression is indeed on-target, and they show that RT cells are selectively dependent upon the DCAF function of PHIP. These results have significantly strengthened this manuscript, so we thank the reviewer again for raising these important points and proposing such informative experiments.

Fig. 3A-B; Given that DDB1 and CUL4 are shared across many CRL4 substrate receptors on chromatin, their broad chromatin loss upon siPHIP is surprising and could reflect off-target effects. **Please include PHIP re-expression rescue to restore chromatin-bound DDB1/CUL4** and, if possible, acute auxin-degron or dTAG approaches to minimize adaptation.

With respect to the use of degrader technology, we have devoted a significant amount of effort to attempting this. Our laboratory has successfully used these systems (both auxin-inducible and dTAG approaches) to study other genes^{19,20}. However, despite extensive efforts, we were unsuccessful in developing similar tools to study PHIP, presumably because the tags interfered with PHIP function.

To provide some detail, we first attempted to exogenously express dTAG-PHIP in RT cells. Constitutive overexpression of dTAG-HA-PHIP (N-term tagged) and PHIP-HA-dTAG (C-term tagged) was poorly tolerated by G401, as was reflected by extremely low expression in pools of edited cells. Nevertheless, we attempted to knockout endogenous PHIP in populations of dTAG-PHIP-expressing cells and to isolate individual clones that expressed only dTAG-PHIP. After screening more than a thousand clones, we identified only a single clone that contained out-of-frame mutations in both alleles of the endogenous PHIP gene. However, these cells expressed only low levels of dTAG-PHIP, and they had severe proliferation defects when compared to normal G401 cells, making them unsuitable for culture and downstream assays. Subsequently, we attempted an alternative approach of knocking an HA-dTAG into the endogenous PHIP locus. Again, we screened more than a thousand clones and identified only one successfully edited clone. Although sequencing confirmed that the HA-dTAG sequence had been successfully inserted into both alleles of the PHIP gene, this clone again displayed severely impaired growth and low levels compared to normal G401 cells. These results demonstrate that putting this tag on either end of PHIP substantially impairs its function, thereby precluding the requested testing.

Accordingly, we pursued an alternate approach to address the query, and this proved successful. We asked how chromatin binding of CRL4 subunits DDB1 and CUL4A was affected by overexpressing full-length PHIP or an H-box mutant of PHIP that failed to interact with CRL4¹⁸. When full-length PHIP was overexpressed, DDB1 and CUL4A accumulated on chromatin (lane 2), an effect not seen when the H-box mutant of PHIP was overexpressed (lane 3). Similarly, knockdown of PHIP impaired CRL4 binding to chromatin (lane 4), an effect that was rescued via the expression of full-length PHIP (lane 5). Consistent with a direct effect, expression of the H-box mutant of PHIP did not rescue. These results are now included in Figure 3D and below.

Fig. 3C; Validate CHD4/MBD3/MTA2/HDAC1/RBBP4 IP-MS interactions by IP-immunoblot. Given the peptide abundance reported, these confirmations should be straightforward.

We have now performed Co-Ips that validated the interactions between PHIP and NuRD subunits in RT cells. The results are shown in Extended Data Figure 3J. PHIP pulldown also enriched NuRD subunits in 293T cells used for ubiquitination assays (see Figure 3G).

Fig. 3D; Clarify the assay: it appears PHIP-FLAG immunocomplexes were incubated with E1/E2 and endogenous CHD4/MBD3/MTA2 ubiquitination was assessed. If that is the case, provide evidence that the CHD4 smear represents polyubiquitination.

Thank you for the opportunity to clarify the results of this experiment. Your interpretation is correct; we were assessing whether FLAG-enriched PHIP could ubiquitinate co-purified NuRD complexes. To determine whether the smear truly represents polyubiquitination, we have now repeated the assay and have added a control condition in which ubiquitin was not added. When ubiquitin was excluded from the assay, we no longer observed a size shift in NuRD subunits, confirming that the smears do indeed represent polyubiquitination. These results are now included in Figure 3G.

Also, reconstitution of fully defined *in vitro* ubiquitination with recombinant CRL4-PHIP and recombinant CHD4 (and other NuRD subunits as controls) to test direct substrate status is required.

Based on your feedback, we have performed additional experiments that support the on-target activity of PHIP in targeting CHD4. We found that overexpressing full-length PHIP enhanced CHD4 turnover on chromatin, whereas overexpressing a mutant version of PHIP that cannot interact with CRL4 did not. These data are now included in Figure 4B.

The experiment that you have proposed is technically challenging and potentially unfeasible for several reasons. The fully synthetic system that you have requested would first require recombinant, catalytically active CRL4-PHIP complexes. This would require the purification and assembly of PHIP and of all the CRL4 subunits. Because PHIP has three reader domains, it would probably need to be assembled on a nucleosome with the correct modifications (H3K4me1/2/3, H3K14ac, H4K12ac) to prevent precipitation. CRL4 is not active unless it is NEDDylated by the DCN complex, so that complex would also need to be purified to add NEDD8 to activate the complex *in vitro*. Because of the technical complexity of generating recombinant and activated CRL4-PHIP complexes, we instead investigated your question by asking whether FLAG-enriched CRL4-PHIP could also polyubiquitinate recombinant NuRD subunits that were added to the assay (CHD4 and 6XHis-HDAC1). The results of this experiment are now included in Extended Data Figure 3I.

When recombinant CHD4 was added (center), we observed additional smearing, suggesting that PHIP can ubiquitinate both co-purified and recombinant CHD4. When recombinant 6XHis-HDAC1 was added (right), we observed ubiquitination of only the co-purified HDAC1, not the tagged recombinant HDAC1 that runs at a higher molecular weight. This suggests that PHIP can directly recognize and ubiquitinate CHD4, indicating that it is a direct substrate. In contrast, only co-purified HDAC1 could be ubiquitinated by PHIP, suggesting that PHIP cannot directly recognize this subunit and may have ubiquitin ligase activity towards it only when it is assembled into NuRD complexes. We speculate that this is potentially useful, as HDAC1 is a member of several other chromatin regulatory complexes.

Based on data in Fig. 3D, it is assumed that CHD4 is the direct target of PHIP. Can the authors determine whether CHD4 is the direct PHIP interactor within the NuRD using *in vitro* binding assay?

We appreciate this feedback and the opportunity to clarify this topic. Our goal was to communicate that our data support a model whereby CRL4–PHIP interacts with the NuRD complex at specific loci (active promoters) and can ubiquitinate certain subunits. We did not intend to suggest that PHIP binds to the NuRD complex exclusively through CHD4, although we agree that this is an interesting question. However, we are concerned with the feasibility of answering this question by using *in vitro* binding assays.

NuRD complexes are large and heterogeneous, with multiple paralogs able to fill the position of each of the seven subunits that make up the ~1-MDa complex. Both the size and the heterogeneity of NuRD complexes would make *in vitro* binding assays challenging. We are also concerned that recombinant PHIP may be a poor representation of endogenous PHIP in *in vitro* binding assays because its three reader domains will not be bound to nucleosomes. Knowing that PHIP localizes to very specific sites on chromatin, it is plausible that PHIP recognizes only those NuRD complexes that are also bound to chromatin, and even then PHIP co-localizes to only a minority of NuRD sites, suggesting that other factors are at play (e.g., post-translational modifications of NuRD and/or co-bound proteins).

Nevertheless, our new results from the ubiquitination assays using recombinant NuRD subunits (described above and shown in Extended Data Figure 3I) suggest that FLAG-enriched PHIP can directly recognize and ubiquitinate recombinant CHD4.

We were then curious to know if CHD4 was necessary for the recognition of NuRD by PHIP and whether PHIP could bind to the complex through other subunits. Accordingly, we performed co-IPs to determine whether PHIP still bound to NuRD when CHD4 or another NuRD subunit (RBBP4) was absent. We found that PHIP could still co-immunoprecipitate NuRD subunits after either subunit was knocked down, which suggests that PHIP does not recognize NuRD complexes exclusively through CHD4 or RBBP4. These results are now shown in Extended Data Figure 3J.

Fig. 3F; PHIP depletion reduces CHD4/HDAC1/2/RBBP4/MBD3 in the NuRD co-eluting fractions. If PHIP promotes CHD4 degradation, one might expect increased CHD4, not reduced signal in the NuRD fractions.

The reviewer makes an important observation, which we had not previously noted. Examining this, in the original figure, we note that the input lanes demonstrated that uneven amounts of nuclear lysate were loaded into the glycerol gradient columns between the control and PHIP-knockdown conditions. Therefore, we repeated the assay to ensure even loading. In this case, there was no longer a reduced abundance of NuRD subunits after PHIP knockdown. These results can now be seen in Figure 3I.

We do still observe that CRL4 subunits (DDB1 and CUL4) no longer migrated with fully assembled NuRD complexes when PHIP was absent. We apologize for the imperfect loading in the original figure and thank the Reviewer for bringing it to our attention.

Fig. 4A, 4C; PHIP overexpression causes only modest decreases in chromatin-bound CHD4/NuRD, and the reported log₂ fold-changes on H4ac sites upon siPHIP are only ~0.2-0.3; small effects for a primary degradation mechanism.

We thank the Reviewer for the opportunity to clarify our results. Because PHIP co-localizes with NuRD complexes at only a minority of NuRD sites (shown in Figure 4C), we expected that many chromatin-bound NuRD complexes would be unaffected by PHIP loss/overexpression (i.e., those at sites at which PHIP was not co-bound). Our results support this, as CHD4 accumulation was observed at PHIP co-

bound sites after PHIP knockdown, whereas CHD4 binding was unchanged at sites where PHIP was not robustly co-bound (see Figure 4F).

We agree that the reductions in H4ac (and other acetylated residues) were modest as detected by ChIP-seq. Orthogonal approaches showed more substantial losses of histone acetylation after PHIP inactivation (by Western blot in Figure 2E, by unbiased histone-MS in Extended Data Figure 2H and I). To examine the on-target activity of PHIP in controlling histone acetylation, we performed H3K27ac ChIP-seq after overexpression, knockdown, or genetic rescue of PHIP, and we present these results in Figure 2G. These findings demonstrate that control of histone acetylation by PHIP is a direct effect.

G

H3K27ac at lost peaks after PHIP KD
LFC < -1, n = 22193

Can the authors test pathway dependence with proteasome inhibition (e.g., MG132/bortezomib) and neddylation inhibition (MLN4924/pevonedistat) to determine whether the observed changes are proteasome- and cullin-dependent.

We thank the reviewer for this insightful suggestion, and we have now performed the requested experiments.

We first tested whether proteasome inhibition could rescue NuRD occupancy after PHIP overexpression. Treatment with the proteasome inhibitor MG132 did not restore NuRD abundance on chromatin under these conditions (Extended Data Fig. 4B).

However, several prior studies have demonstrated that even low levels of MG132 treatment can induce autophagy as a secondary mechanism of degradation (PMID: 28674081, 30647455, 32092124). Consequently, the failure to rescue could be a consequence of compensatory autophagic degradation of chromatin-associated proteins.

As suggested by the reviewer, to address pathway dependence by using a complementary approach, we next tested whether inhibiting cullin activity could rescue NuRD binding. In contrast to MG132 treatment, treatment with MLN4924 (pevonedistat), which inhibits the NEDD8-activating enzyme and thereby prevents activation of cullin-RING ligases (including CUL4), robustly blocked PHIP-mediated suppression of NuRD on chromatin (Extended Data Fig. 4A).

Importantly, we now show that genetic perturbations of PHIP that disrupted its interaction with CRL4 similarly abolished PHIP-dependent NuRD suppression (Fig. 4B).

Together, these pharmacological and genetic data demonstrate that the regulation of NuRD by PHIP is cullin-dependent and that the activity of PHIP results in the loss of NuRD from chromatin. With respect to these findings, it is noteworthy that ubiquitination can trigger substrate removal from chromatin independently of degradation in several ways. Segregases, such as p97/VCP, can recognize and extract ubiquitinated substrates from chromatin, resulting in inactivation that is independent of proteasomal turnover (PMID: 27086594). Additionally, a study demonstrating that post-translational SUMOylation of NuRD subunits can trigger their removal from chromatin without inducing degradation (PMID: 27068747) suggests that post-translational modification of NuRD alone can interfere with its assembly onto chromatin.

Because the functional opposition of NuRD by PHIP is clear and the downstream consequence of PHIP-mediated NuRD suppression (impaired histone deacetylation and activation of pro-proliferative transcriptional programs) will be identical regardless of whether NuRD is degraded or simply evicted from chromatin, our conclusions regarding PHIP-dependent regulation of NuRD are fully supported by the current data. We have updated our text in several locations to specify that PHIP “ubiquitinates and suppresses” NuRD complexes, rather than “ubiquitinates and degrades” them. We view the precise fate of ubiquitinated NuRD as a potential topic for future investigation, but not one that alters the central conclusions of this study.

We again thank the reviewer for prompting these experiments, which have substantially strengthened our mechanistic understanding of PHIP-mediated NuRD regulation.

Fig. 6B, 6D; Are the phenotypes caused by PHIP loss rescued by co-depletion of CHD4 (or stabilized CHD4)? Epistasis would strongly support CHD4 as the relevant substrate/effector in this context.

As most CHD4/NuRD binding sites do not have co-binding of PHIP (see Figure 4C), the effects of CHD4 loss will extend beyond the sites where PHIP is suppressing its activity. With respect to this, analysis of CHD4 loss in DepMap shows that it is a common essential gene across almost all cell lines (a CHRONOS scores < -0.5 indicates robust dependence upon a gene).

Therefore, although the question is conceptually interesting, this is not testable in practice because most NuRD binding sites do not co-localize with PHIP and CHD4 is a pan-essential gene.

Minor Concerns:

Lines 56-58: “SMARB1” should be SMARCB1. Also, the blanket statement that SMARCB1 and SMARCA4 are “bona fide tumor suppressors” is imprecise; SMARCA4 can act as an oncogene in certain contexts (e.g., breast, prostate, colorectal; PMID: 36769189).

We have corrected the typo. The reviewer is correct that whereas genetic inactivation of SMARCB1 and SMARCA4 has been shown to cause cancer in both mouse models and germline inactivation of either predisposes individuals to cancer, they have also been implicated as essential in certain other cancers. Consequently, we have now added the clarification that SMARCB1 and SMARCA4 “exhibit bona fide tumor suppressor activity, as germline heterozygous mutations are associated with cancer predisposition in humans and knockout in mice results in tumor formation.”

Many main-figure analyses use HA-PHIP ChIP-seq from G401 cells overexpressing HA-PHIP. This should be explicitly noted in the main text (not only legends), with rationale for choosing HA-PHIP over endogenous PHIP and a discussion of potential overexpression confounders.

Thank you for raising this important point. In our early experiments, we struggled to get ChIP-seq to work for endogenous PHIP, so we used HA-PHIP. We were subsequently able to get ChIP-seq to work for endogenous PHIP. Given the reviewer’s concerns, we have repeated all relevant analyses using ChIP-seq for endogenous PHIP, and we have replaced the HA-PHIP data with endogenous PHIP data in all heatmaps. The interpretations have not changed.

- 1 Wang, X. *et al.* SMARCB1-mediated SWI/SNF complex function is essential for enhancer regulation. *Nat Genet* **49**, 289-295 (2017). <https://doi.org/10.1038/ng.3746>
- 2 Malone, H. A. & Roberts, C. W. M. Chromatin remodellers as therapeutic targets. *Nat Rev Drug Discov* **23**, 661-681 (2024). <https://doi.org/10.1038/s41573-024-00978-5>
- 3 Howard, T. P. *et al.* MDM2 and MDM4 are therapeutic vulnerabilities in malignant rhabdoid tumors. *Cancer Res* **79**, 2404-2414 (2019). <https://doi.org/10.1158/0008-5472.Can-18-3066>
- 4 Michel, B. C. *et al.* A non-canonical SWI/SNF complex is a synthetic lethal target in cancers driven by BAF complex perturbation. *Nat Cell Biol* **20**, 1410-1420 (2018). <https://doi.org/10.1038/s41556-018-0221-1>
- 5 Wang, X. *et al.* BRD9 defines a SWI/SNF sub-complex and constitutes a specific vulnerability in malignant rhabdoid tumors. *Nat Commun* **10**, 1881 (2019). <https://doi.org/10.1038/s41467-019-09891-7>
- 6 Gounder, M. *et al.* Tazemetostat in advanced epithelioid sarcoma with loss of INI1/SMARCB1: an international, open-label, phase 2 basket study. *Lancet Oncol* **21**, 1423-1432 (2020). [https://doi.org/10.1016/s1470-2045\(20\)30451-4](https://doi.org/10.1016/s1470-2045(20)30451-4)
- 7 Morris, S. A. *et al.* Overlapping chromatin-remodeling systems collaborate genome wide at dynamic chromatin transitions. *Nat Struct Mol Biol* **21**, 73-81 (2014). <https://doi.org/10.1038/nsmb.2718>
- 8 Yildirim, O. *et al.* Mbd3/NURD complex regulates expression of 5-hydroxymethylcytosine marked genes in embryonic stem cells. *Cell* **147**, 1498-1510 (2011). <https://doi.org/10.1016/j.cell.2011.11.054>
- 9 de Dieuleveult, M. *et al.* Genome-wide nucleosome specificity and function of chromatin remodellers in ES cells. *Nature* **530**, 113-116 (2016). <https://doi.org/10.1038/nature16505>
- 10 Low, J. K. K. *et al.* The nucleosome remodeling and deacetylase complex has an asymmetric, dynamic, and modular architecture. *Cell Reports* **33**, 108450 (2020). <https://doi.org/https://doi.org/10.1016/j.celrep.2020.108450>
- 11 Low, J. K. K. *et al.* CHD4 is a peripheral component of the nucleosome remodeling and deacetylase complex. *J Biol Chem* **291**, 15853-15866 (2016). <https://doi.org/https://doi.org/10.1074/jbc.M115.707018>
- 12 Zhang, W. *et al.* The nucleosome remodeling and deacetylase complex NuRD is built from preformed catalytically active sub-modules. *J Mol Biol* **428**, 2931-2942 (2016). <https://doi.org/10.1016/j.jmb.2016.04.025>
- 13 Guschin, D., Wade, P. A., Kikyo, N. & Wolffe, A. P. ATP-dependent histone octamer mobilization and histone deacetylation mediated by the Mi-2 chromatin remodeling complex. *Biochemistry* **39**, 5238-5245 (2000). <https://doi.org/10.1021/bi000421t>
- 14 Tong, J. K., Hassig, C. A., Schnitzler, G. R., Kingston, R. E. & Schreiber, S. L. Chromatin deacetylation by an ATP-dependent nucleosome remodeling complex. *Nature* **395**, 917-921 (1998). <https://doi.org/10.1038/27699>
- 15 Xue, Y. *et al.* NURD, a novel complex with both ATP-dependent chromatin-remodeling and histone deacetylase activities. *Mol Cell* **2**, 851-861 (1998). [https://doi.org/10.1016/s1097-2765\(00\)80299-3](https://doi.org/10.1016/s1097-2765(00)80299-3)
- 16 Nakayama, R. T. *et al.* SMARCB1 is required for widespread BAF complex-mediated activation of enhancers and bivalent promoters. *Nat Genet* **49**, 1613-1623 (2017). <https://doi.org/10.1038/ng.3958>
- 17 Loughran, A. J. *et al.* Rapid and robust validation of pooled CRISPR knockout screens using CelFi. *Sci Rep* **15**, 13358 (2025). <https://doi.org/10.1038/s41598-025-96095-3>

- 18 Kim, D.-K., Redon, C. E., Aladjem, M. I., Kim, H. K. & Jang, S.-M. Molecular double clips within RepID WD40 domain control chromatin binding and CRL4-substrate assembly. *Biochem Biophys Res Commun* **567**, 208-214 (2021). <https://doi.org/10.1016/j.bbrc.2021.06.047>
- 19 Radko-Juettner, S. *et al.* Targeting DCAF5 suppresses *SMARCB1*-mutant cancer by stabilizing SWI/SNF. *Nature* **628**, 442-449 (2024). <https://doi.org/10.1038/s41586-024-07250-1>
- 20 Zhu, Z. *et al.* Mitotic bookmarking by SWI/SNF subunits. *Nature* **618**, 180-187 (2023). <https://doi.org/10.1038/s41586-023-06085-6>